# Engineering a scalable and orthogonal platform for synthetic communication in mammalian cells

Anna-Maria Makri Pistikou [1,2,3,4,9], Glenn A. O. Cremers[1,2,3,9],
Bryan L. Nathalia [1,2,3,4,9], Theodorus J. Meuleman[1,2,3,4,5], Bas W. A. Bögels [1,2,3],
Bruno V. Eijkens[3,4], Anne de Dreu [1], Maarten T. H. Bezembinder[1,2,3,4],
Oscar M. J. A. Stassen [4], Carlijn C. V. Bouten [2,4], Maarten Merkx [1,2],
Roman Jerala [6,7] & Tom F. A. de Greef [1,2,3,4,5,8] ✉

The rational design and implementation of synthetic mammalian communication systems can unravel fundamental design principles of cell communication circuits and offer a framework for engineering of designer cell consortia with potential applications in cell therapeutics. Here, we develop the foundations of an orthogonal, and scalable mammalian synthetic communication platform that exploits the programmability of synthetic receptors and selective affinity and tunability of diffusing coiled-coil peptides. Leveraging the ability of coiled-coils to exclusively bind to a cognate receptor, we demonstrate orthogonal receptor activation and Boolean logic operations at the receptor level. We show intercellular communication based on synthetic receptors and secreted multidomain coiled-coils and demonstrate a three-cell population system that can perform AND gate logic. Finally, we show CC-GEMS receptor-dependent therapeutic protein expression. Our work provides a modular and scalable framework for the engineering of complex cell consortia, with the potential to expand the aptitude of cell therapeutics and diagnostics.

The ability to engineer novel functions in mammalian cells has become key in biomedical research, revolutionizing the field of cell-based diagnostics and therapeutics[1–4]. In particular, the introduction of synthetic receptors[5,6] has enabled the engineering of designer cells that, due to their sensing and actuating capabilities, can detect and correct disease state[7–16]. To further improve on the specificity of such approaches, multilayered circuits for combinatorial detection of multiple biomarkers need to be introduced[17–19]. Nonetheless, large genetic circuits are challenging to implement in monocultures of cells, as they are met with limitations due to the burden introduced by resource sharing[20–22]. In addition, delivery of multiple genetic constructs can be further restricted due to the limited effective packaging capacity of the vector[23]. To fully unlock the potential of such technology, engineered cells need to communicate reciprocally and aptly process information in a way that resembles specialized cell consortia of the human immune system. The human immune system is a compelling model

[1]Laboratory of Chemical Biology, Department of Biomedical Engineering, Eindhoven University of Technology, Eindhoven, The Netherlands. [2]Institute for Complex Molecular Systems, Department of Biomedical Engineering, Eindhoven University of Technology, Eindhoven, The Netherlands. [3]Computational Biology Group, Department of Biomedical Engineering, Eindhoven University of Technology, Eindhoven, The Netherlands. [4]Laboratory for Cell and Tissue Engineering, Department of Biomedical Engineering, Eindhoven University of Technology, Eindhoven, The Netherlands. [5]Center for Living Technologies, Eindhoven-Wageningen-Utrecht Alliance, Utrecht, The Netherlands. [6]Department of Synthetic Biology and Immunology, National Institute of Chemistry, Ljubljana, Slovenia. [7]EN-FIST Centre of Excellence, Ljubljana, Slovenia. [8]Institute for Molecules and Materials, Radboud University, Nijmegen, The Netherlands. [9]These authors contributed equally: Anna-Maria Makri Pistikou, Glenn A. O. Cremers, Bryan L. Nathalia. ✉e-mail: t.f.a.d.greef@tue.nl

target, since it has the exceptional capacity to sense and respond to a multitude of diffusing signals, perform distributed computing, and leverage specialized cell types to address challenges[24]. An envisioned mammalian synthetic communication network has the potential to reduce burden due to resource competition on individual cell populations by permitting distributed information processing, leading to advanced functions[25,26]. Such a system can enable population control as well as allow for coordinated responses of therapeutic cells, in a manner similar to previously developed synthetic prokaryotic communication systems[27–29]. To realize intercellular communication and enable programmable control of neighboring cell responses, ideally cells are equipped with the capacity to sense orthogonal, soluble stimuli while at the same time producing user-defined responses. Synthetic, intercellular, juxtracrine communication between mammalian cells has been previously achieved utilizing synthetic receptors[12,15,30] that can perform logic operations[17–19,31,32] based on cell-to-cell contact. Efforts to introduce diffusion-based, intercellular synthetic communication in mammalian cells include the design of mammalian circuitry deploying repurposed small molecules[33–38], in addition to approaches that employ directed evolution of naturally occurring proteins[39–41]. However, directed evolution of natural proteins consists of a labor-intensive method to render input signals orthogonal from each other,

while in addition such an approach lacks the inherent ability to institute Boolean logic. Furthermore, although a set of chimeric receptors have been engineered using an orthogonal interleukin 2 (IL-2) extracellular domain (ECD) to the γ-chain ($\gamma_c$) of a set of cytokines[41], the elicited response is limited to the corresponding $\gamma_c$ cytokine-related signal. Conversely, approaches utilizing small molecules have managed to engineer mammalian cells so they are capable of complex biocomputations[42]. However, these approaches lack in scalability due to the use of small molecules as signal initiators and cannot perform Boolean logic operations at the receptor level. While these advancements have arguably managed to construct synthetic communication between mammalian cells, there is currently no available platform for synthetic communication, utilizing diffusible ligands, that is inherently scalable, orthogonal to the native cell, and able to perform logic operations at the receptor level.

In this work, we engineer a scalable and orthogonal synthetic communication platform for mammalian cells based on designed diffusible ligands. In detail, we functionalised erythropoietin receptor (EpoR) domains from the Generalized Extracellular Molecule Sensor (GEMS) platform[10] with coiled-coil peptides (CC) from the NICP set[43,44], that can mutually and orthogonally bind to each other (Fig. 1a). Programmable design of large sets of CC heterodimers has been

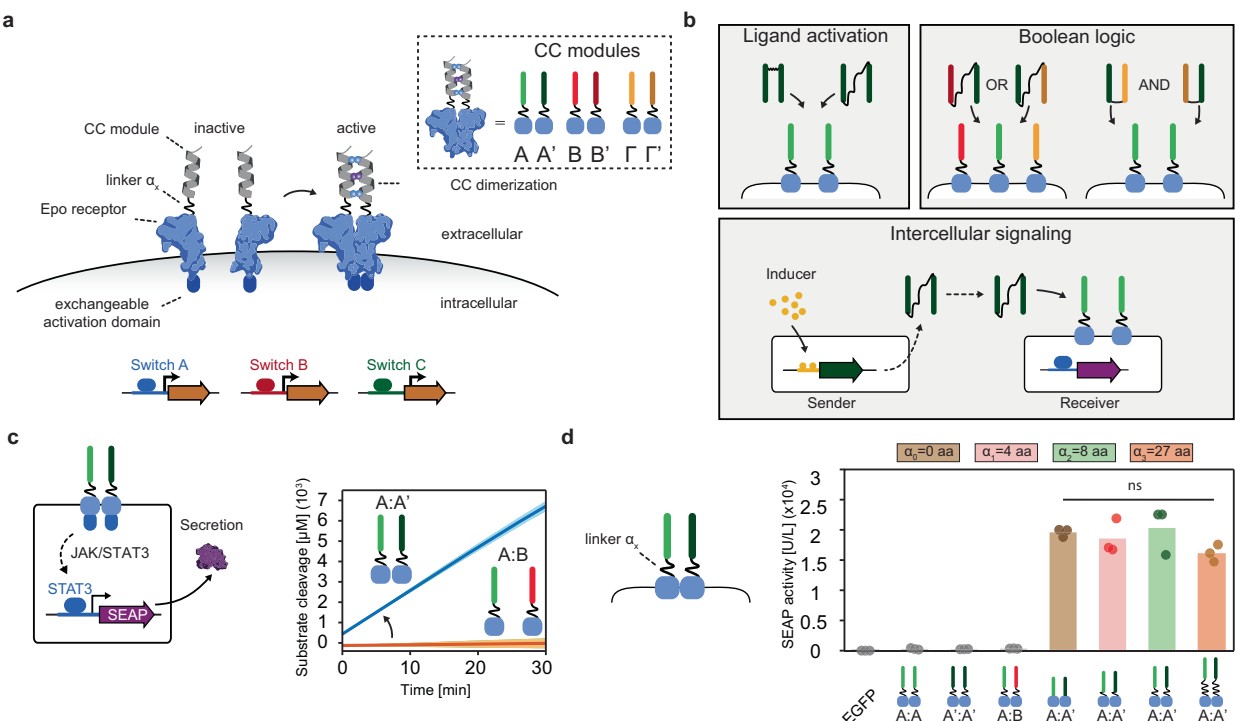

**Fig. 1 | Design elements and general overview of a coiled-coil (CC)- functionalised GEMS synthetic communication platform for mammalian cells.**
**a** Schematic representation of CC-functionalised GEMS receptor resulting in activation of target genes, upon dimerization. CC peptides are N-terminally fused, through linker $\alpha_x$ to the extracellular and transmembrane domains of the erythropoietin receptor (EpoR) cluster that can induce transgene expression following activation of an intracellular signaling pathway. Complementary CC modules are indicated as A:A', B:B', and Γ:Γ' in the inset. Downstream receptor signaling can be modulated by exchanging the intracellular activation domains (switch A, B, and C) and output can be expressed by replacing the reporter gene to a transgene of choice. **b** Receptor activation can be achieved by soluble ditopic CC ligands the properties of which can be utilized to achieve Boolean logic gate operations (AND/OR). Intercellular communication can be achieved by engineering sender cells that express the ligand under the control of an ON/OFF switch and a receiver cell population expressing the cognate receptor and reporter gene. **c** Schematic overview of the reporter system to monitor receptor activation using JAK/STAT

intracellular signaling. Each receptor monomer bears a cognate CC (A and A' with linker $\alpha_2$) that can cause the receptor monomers to heterodimerize. Phosphorylated STAT3 results in the production of the reporter protein: human placental secreted alkaline phosphatase (SEAP) that catalyzes the hydrolysis of a p-Nitrophenylphosphate (pNPP). Substrate conversion is observed when the cognate receptors A:A' are expressed on the cell surface, while non-cognate pairs (A:B) do not result in receptor activation. The experiment is performed in independent triplicates; solid line indicates mean and shaded area standard deviation. **d** SEAP activity [U/L] in HEK293T cells transiently transfected with receptor pairs with varied linker lengths $\alpha_x$ of zero, eight, and 27 amino acids (aa; GS or GSS repeats). SEAP activity was measured 48 h following transfection. Bars indicate mean activity; individual data points represent independent triplicates, performed on the same day. Significance (one-way ANOVA with Tukey's multiple comparison test) is noted above the bars (see Supplementary Table S4). Source data are provided as a Source Data file.

previously achieved[45], providing the platform with potential scalability. The similarity in peptide size and conformation between individual CC interaction domains generalizes the receptor activation mechanism and allows for the introduction of different CC modules on the EpoRs without having to re-engineer the system for each orthogonal ligand-receptor pair. To enable synthetic communication between populations of mammalian cells, we design a small ubiquitin-like modifier (SUMO) tag-fused CC dipeptide that can be secreted by mammalian cells. We first demonstrate that receptors carrying heterodimeric, complementary CC peptides result in robust receptor activation upon expression. The design and engineering of ditopic CC peptide ligands allows for cognate receptor activation by the addition of the ditopic ligands to cell culture, for receptors utilizing two distinct pathways. The present work exemplifies the scalability of our platform by demonstrating synthetic receptor activation by three unique CC pairs. In addition, we prove that mammalian cells expressing a combination of synthetic receptors can perform two-input logic (AND and OR gate operations) (see Fig. 1b). Although CC-mediated Boolean gates have been engineered in mammalian cells at the intracellular level[46], they have not been used to control receptor activation and enable intercellular communication. Next, we achieve synthetic intercellular communication in a minimal system composed of a sender population that secretes a synthetic dipeptide ligand and a receiver population expressing the cognate receptor and reporter. Establishing an inducible intercellular communication system allows the engineering of a three-cell-population system that can perform distributed AND gate operations. Finally, to show a potential application of the CC-GEMS platform, we demonstrate therapeutic protein expression mediated by CC-GEMS receptor activation. Our design offers a customizable and extensible framework facilitating the engineering of intricate consortia of designer cells that can process information and autonomously respond to stimuli. The present work holds the potential to expand the aptitude of cell therapeutics and diagnostics, and also offers a platform to study fundamental principles of mammalian cell–cell communication circuits.

## Results

### Design principles of a coiled-coil-functionalised GEMS synthetic communication platform

To actualize synthetic communication in mammalian cells, we focused our attention on the previously established Generalized Extracellular Molecule Sensor (GEMS) platform[10], which allows for customized input and output. GEMS enables the engineering of modular receptor designs that integrate user-defined, soluble ligand sensing through the introduction of a variety of engineered ligand-binding extracellular domains to transgene expression via the rewiring of distinct endogenous signaling pathways[10,47–49]. In detail, GEMS receptors consist of a standard transmembrane scaffold, the erythropoietin receptor (EpoR) that is engineered to be inert to erythropoietin, fused to intracellular signaling domains, derived from the cytokine receptor chain interleukin 6 receptor B (IL-6RB), the fibroblast growth factor receptor 1 (FGFR1), or the vascular endothelial growth factor receptor 2 (VEGFR2). GEMS dimerization leads to downstream signaling by activating the Janus kinase/signal transducer and activator of transcription (JAK/STAT) pathway (induced by IL-6RB), the mitogen-activated protein kinase (MAPK) pathway (induced by FGFR1), the phosphatidylinositol 3-kinase/protein kinase B (PI3K/Akt) pathway, or the phospholipase C gamma (PLCG) pathway (both PI3K/Akt and PLCG induced by VEGFR2). Transgene expression is achieved by rerouting the abovementioned pathways using minimal promoters engineered to be responsive to specific pathways[10]. We functionalised the extracellular part of the EpoR transmembrane scaffold of the GEMS receptor with coiled-coil peptides (CC), that possess the ability to mutually and orthogonally bind to each other[43,44] (Fig. 1a). Receptor heterodimerization is thus induced by cognate CC pairing; A binds

exclusively to A', B to B' and Γ to Γ' (see Supplementary Table S1). Each CC is genetically fused to an EpoR domain through linker $\alpha_x$ (see Supplementary Table S2 and Methods). Non-cognate CC-GEMS receptors should not activate intracellular signaling, while cognate CC pairing is expected to result in transgene expression due to receptor heterodimerization.

To monitor CC-induced receptor activation, HEK293T cells were transiently transfected with CC-GEMS receptors utilizing the JAK-STAT pathway (Supplementary Fig. S1), as well as *STAT3*, and the reporter gene *STAT3*-induced secreted alkaline phosphatase (*SEAP*; see "Methods" and Supplementary Table S3). Following receptor activation, *STAT3* is phosphorylated and becomes a transcription factor for the *SEAP* reporter gene. *SEAP* secretion was quantified in the cell supernatant using a colorimetric assay that reports substrate conversion (see "Methods", Fig. 1c and Supplementary Fig. S2). When cells were co-transfected with A-type$_{JAK/STAT}$ and A'-type$_{JAK/STAT}$ cognate receptors with linker $\alpha_2$ (4× glycine–serine (GS) repeats), a sharp increase in substrate conversion was observed, indicating receptor activation. As hypothesized, non-cognate receptor pairs (A:B) did not result in receptor activation (see Fig. 1c).

Next, we investigated whether receptor activation is dependent on the linker length between the CC domain and the transmembrane scaffold. We designed constructs with varying linker lengths $\alpha_x$ (see Supplementary Table S2) of zero, four ($\alpha_1$; (GS)$_2$), eight ($\alpha_2$; (GS)$_4$) and 27 aa ($\alpha_3$; (GSS)$_9$), tethering the CC and EpoR domains. Subsequently, HEK293T cells were transfected with a range of CC-GEMS$_{JAK/STAT}$ pairs, *STAT3* and reporter gene *SEAP*. SEAP activity was quantified in the cell supernatant (see "Methods"). To determine background alkaline phosphatase activity and assess transfection efficiency, HEK293T cells were transiently transfected with EGFP. The transfection efficiency was estimated to be 62.1% (see Supplementary Fig. S3) and the background alkaline phosphatase activity was negligible (see Fig. 1d). For the four different linker lengths used in this study, our data reveals that receptor activation is not critically dependent on linker length as all cognate heterodimer receptor pairs (A:A') resulted in robust SEAP expression (Fig. 1d and Supplementary Table S4 for statistical analysis), with no apparent linker-dependent effect. However, since linker-length dependence has been reported in other synthetic receptors[50], further research is needed to investigate a broader range of linker lengths and its influence on CC-GEMS receptor activation. All non-cognate receptor pairs (A:A, A':A', and A:B with linker $\alpha_2$) showed negligible receptor activation, verifying the orthogonality of CC pairs, and demonstrating the programmability of our approach. Collectively, these results indicate that CC interactions can be used as a robust, programmable tool for synthetic receptor activation.

### Design of soluble ditopic CC ligands to activate CC-GEMS receptors

Having established that cognate CC pairs are able to induce receptor heterodimerization, we next aimed to assess if CC-GEMS receptor dimerization and activation can be achieved by soluble, synthetic, CC ligand dipeptides. Considering the parallel orientation of bound CC cognate pairs[43], we engineered A'-A' dipeptides that are N-termini linked (Fig. 2a). To that end, we modified the A' peptide sequence to include a single cysteine at the N-terminus (see Fig. 2b and Supplementary Fig. S4a, S4b). As recombinant expression of short peptides in *E. coli* can result in protein degradation or the formation of inclusion bodies[51,52], we increased the expression and solubility of the peptide by fusion to a SUMO tag[53,54]. We expressed SUMO-A' fusion protein in *E. coli* (see "Methods" and Supplementary Fig. S5a). Successful expression and purification of the recombinant protein was confirmed, using sodium dodecyl sulfate-polyacrylamide gel electrophoresis (SDS-PAGE; "Methods" and Supplementary Fig. S4c) and subsequent SUMO cleavage and purification resulted in the recovery of the cysteine-functionalized A' peptide (Supplementary Fig. S4d). We employed the

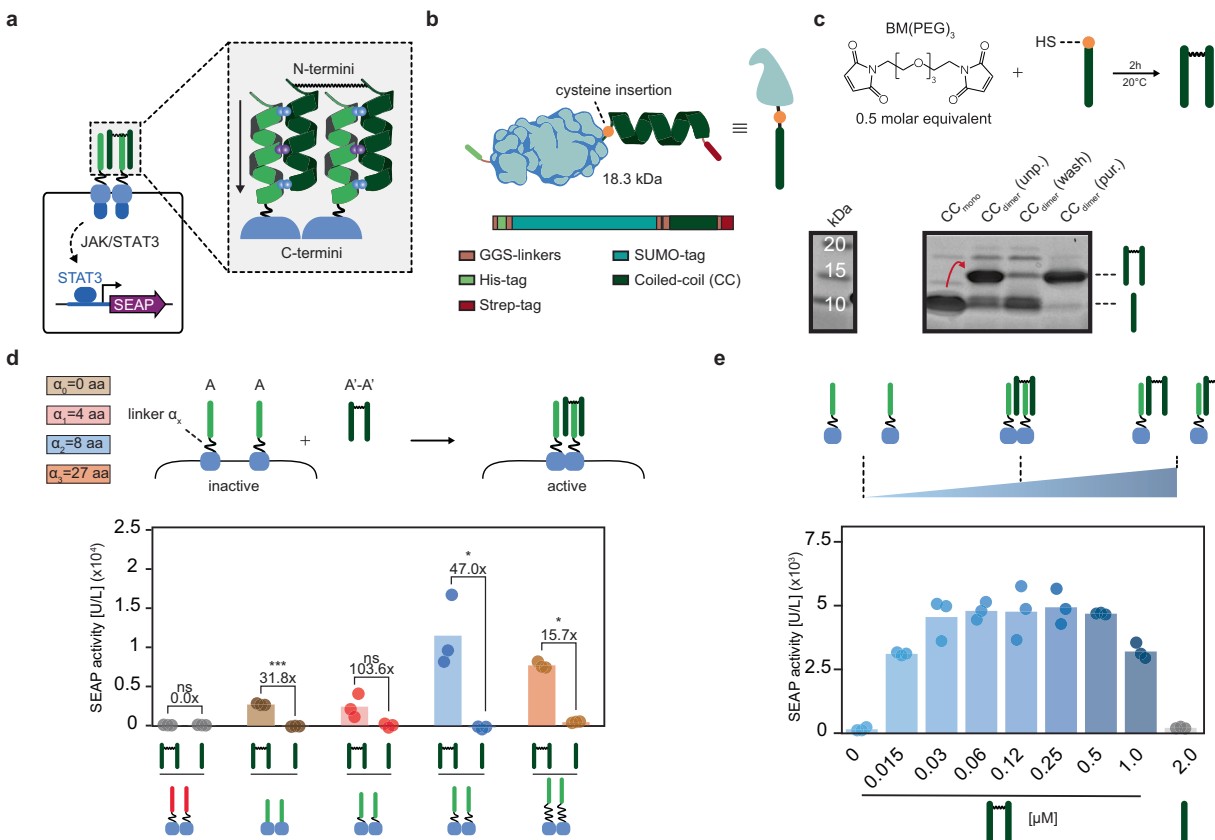

**Fig. 2 | Synthetic, soluble, ditopic CC ligands result in robust receptor activation. a** Schematic overview of the envisioned ligand-receptor interaction. N-termini linkage of two monomeric CCs results in the desired parallel conformation of a bivalent CC ligand for receptor activation. Arrow indicates N- to C- terminus directionality. **b** Schematic representation of the expression strategy of the monomeric CC ligand in *E. coli*. An N-terminus, cleavable small ubiquitin-like modifier (SUMO) tag is fused to a CC ligand, having a single cysteine at the C-terminus. For purification purposes, the fusion protein harbors an N-terminal hexahistidine (His) tag and a C-terminal Strep-tag, connected through a small, flexible glycine–serine (GS) linker (see "Methods" and Supplementary Fig. S4). **c** SDS-PAGE analysis showing the successful engineering of the dipeptide prior to purification (CC$_{dimer}$(unp.). Anion-exchange chromatography results in removal of unreacted CC monomers (CC$_{dimer}$(wash) and recovery of the ditopic CC ligands (CC$_{dimer}$(pur.) (see "Methods" and Supplementary Fig. S4). **d** SEAP activity [U/L] in

HEK293T cells transiently transfected with CC-GEMS receptors (A-type$_{JAK/STAT}$ or B-type$_{JAK/STAT}$) with varied linker lengths $\alpha_x$ of zero, four, eight, and 27 aa (GS or GSS repeats). Following transfection, cells were incubated with either 0.12 μM purified A'-A' dipeptide or 0.24 μM A' monomeric CC and SEAP expression was monitored to assess receptor activation (see "Methods"). Fold change and significance (two-tailed, unpaired *t* test) is noted above bars. ns: $P > 0.05$, $^*P \le 0.05$, $^{**}P \le 0.01$, $^{***}P \le 0.001$ (see Supplementary Table S6). **e** Titration of ditopic ligand A'-A' on HEK293T cells transiently transfected to express the A-type$_{JAK/STAT}$ receptor with $\alpha_2$ linker (8 aa, GS repeats). Cells were incubated with a range of concentrations of ligand A'-A' (0–1 μM) or 2 μM of monomeric A' for 48 h. SEAP activity was monitored to assess receptor activation. Bars indicate mean activity; individual data points represent independent triplicates, performed on the same day. Source data are provided as a Source Data file.

sulfhydryl moiety of the cysteine to synthesize N-termini-linked ditopic A'-A' ligand, using a bifunctional maleimide linker (see "Methods" and Fig. 2c, top). SDS-PAGE gel analysis illustrated the successful formation of A'-A' dipeptides and subsequent purification with anion-exchange chromatography resulted in the removal of unreacted A' monomers (Fig. 2c, bottom). To validate that dipeptide formation occurs due to reaction of sulfhydryl moieties of A' monomers, we incubated A' monomers that were site-specifically mutated to include a glycine in place of cysteine (A'.C4G (v$_1$); Supplementary Table S5 and Supplementary Fig. S6a, S6b) with the bifunctional linker and observed no dipeptide formation (Supplementary Fig. S6c).

After successful synthesis and purification of A'-A' dipeptides, we aimed to evaluate their ability to dimerize and activate A-type$_{JAK/STAT}$ receptors. For this purpose, we transfected HEK293T cells with A-type$_{JAK/STAT}$ receptors, *STAT3* and *SEAP* reporter gene and incubated the transfected cells with A'-A' dipeptides or A' monomeric peptides (A'.C4G (v$_1$); see "Methods" and Fig. 2d). Our results reveal that A'-A' dipeptides can activate A-type$_{JAK/STAT}$ receptors, as evident by an increase in SEAP expression. As expected, no receptor activation was observed for monomeric A'. Both A' monomers and A'-A' dipeptides

failed to activate non-cognate B-type$_{JAK/STAT}$ receptors, as anticipated. To investigate the influence of linker $\alpha_x$ between the A-type$_{JAK/STAT}$ CC and EpoR domain on ligand-dependent receptor activation, we expressed cells with A-type$_{JAK/STAT}$ receptors with no linker, linker $\alpha_1$, $\alpha_2$, or $\alpha_3$ (see Supplementary Table S2), and incubated them with A'-A' dipeptide. We observe a 31.8-fold increase for linker $\alpha_x$ of 0 aa, 103.6-fold increase for linker $\alpha_1$, 47.0-fold increase for linker $\alpha_2$ and a 15.7-fold increase for the longer linker $\alpha_3$, when compared to incubation with monomeric A's. We therefore employed receptors with an $\alpha_2$ linker in further experimental set-ups. Interestingly, we noticed a decrease of approximately 60% SEAP expression for ligand-induced type activation compared to the situation in which cells express cognate A-type$_{JAK/STAT}$ and A'-type$_{JAK/STAT}$ receptor pairs with linker $\alpha_2$ (Supplementary Fig. S7). We hypothesized that these differences in activation levels can be explained by the number of dimerized CC-GEMS receptors. Specifically, when cognate CC-GEMS are concurrently expressed, dimerization can, in principle, already take place within the secretory pathway before CC-GEMS receptors are translocated to the cell membrane. This would result in increased activation levels, compared to CC-GEMS receptors dimerized due to the presence of a

cognate CC dipeptide ligand, since soluble ligands can only activate receptors displayed on the cell membrane. In addition, endogenous receptor dimerization occurs without conformational restriction induced by membrane organization[55] which could result in a different receptor activation mechanism compared to receptor activation on the membrane. As a result, diverse outcomes related to linker-length dependent activation can occur between cognate CC-GEMS receptor pairs that already dimerize in the secretory pathway compared to CC-GEMS receptors that dimerize on the membrane upon addition of a cognate ligand. Furthermore, a cognate CC-GEMS heterodimer, such as A'-type$_{JAK/STAT}$:A-type$_{JAK/STAT}$ receptor pair will result in a different receptor dimer proximity compared to a homodimeric CC-GEMS activated by an external ligand, such as A-type receptor pairs bound to an A'-A' dipeptide. The difference in receptor dimer proximity between a receptor heterodimer and a ligand-activated CC-GEMS should be considered when considering the diverse outcomes of linker length in receptor activation (Figs. 1d and 2d). Ligand-induced receptor dimerization typically results in a bell-shaped dose-response curve that is susceptible to changes in ligand concentration[56]. When titrating A'-A' dipeptides to HEK293T cells transiently transfected with A-type receptors with linker $\alpha_2$, STAT3 and SEAP reporter gene, a bell-shaped dose-response curve with a broad plateau, for concentrations ranging between 0.03 and 0.5 $\mu$M was observed (Fig. 2e).

To further investigate ligand design, we expressed two alternative ditopic A'-A' ligands (see Supplementary Table S5) that contained a small N-terminal linker that included either one ($v_2$ in Supplementary Fig. S6a, S6b, S6d) or two repeats ($v_3$ in Supplementary Fig. S6a, S6b, S6e) of the strong helix-interrupting residues proline and glycine[57]. Experiments reveal that the alternative ditopic A'-A' ligands $v_2$ and $v_3$ result in robust receptor activation in HEK293T cells that are transiently transfected with A-type$_{JAK/STAT}$ receptor, STAT3, and SEAP reporter (Supplementary Fig. S6f and Supplementary Table S4).

These results collectively demonstrate that synthetic, CC ligand dipeptides can activate cognate CC-GEMS receptors.

Having established that synthetic, N-termini linked dipeptides can activate cognate CC-GEMS receptors, resulting in the expression of the reporter gene, we next sought to engineer a ditopic CC ligand that can be expressed and secreted, and can thus facilitate intercellular communication. Accordingly, we envisaged two CC peptides that are genetically fused through a long polypeptide linker ($l_x$), bridging the N-terminus of one CC to the C-terminus of another (see Fig. 3a). The anticipated minimum length that a linker needs to bridge between the C-terminus of one CC domain and the N-terminus of another has been estimated to be approximately 4 nm (calculated distance based on the length of $\alpha$-helix for a single CC domain, see Methods). To span the two CCs, we made use of three linkers, comprising a combination of rigid $\alpha$-helical blocks ((EAAAK)$_6$)[58] and more flexible domains ((SGSSGS)$_3$)[59], namely, $l_1$, $l_2$, and $l_3$ (see "Methods", Fig. 3b and Supplementary Table S7)[60,61]. To engineer A'-A' dipeptides with linker $l_x$, we initially employed a bacterial expression system (see "Methods" and Supplementary Fig. S5b). Successful expression and purification of the SUMO-tagged A'-A' was confirmed using SDS-PAGE and successive SUMO cleavage ("Methods" and Supplementary Fig. S8a) resulted in the recovery of pure A'-A' dipeptides, with linker $l_1$ (Supplementary Fig. S8b), $l_2$ (Supplementary Fig. S8c) and $l_3$ (Supplementary Fig. S8d). To assess the ability of recombinantly expressed A'-A' ditopic peptides connected with a linker $l_x$ to activate cognate CC-GEMS receptors, we transiently transfected HEK293T cells with A-type$_{JAK/STAT}$ receptors with $\alpha_2$, STAT3 and SEAP reporter gene and incubated the transfected cells with the A'-A' ligands with and without the SUMO fusion ("Methods" and Fig. 3c). We observed reporter gene expression for all the cognate ligand-receptor pairs, while the presence of the SUMO tag did not inhibit receptor activation (see Supplementary Table S4 for statistical analysis). A'-A' ligands with a semi-flexible $l_2$ spacer (Fig. 3c, yellow bars) triggered similar receptor activation levels as synthetic A'-

A' ligands (Fig. 3c, blue bar), demonstrating 15.5-fold increase for synthetic A'-A' and 16.6-fold increase for expressed A'-A' with $l_2$, when compared to cells incubated with A' monomers. A'-A' ligands with a shorter $l_1$ linker resulted in even higher receptor activation. More specifically, we observed a 20.7-fold increase for SUMO-tagged A'-A' with $l_1$ and 24.7-fold for A'-A' with $l_1$, when compared to cells incubated with A' monomers. A'-A' ligands with the longest linker $l_3$ resulted in the lowest receptor activation; 8.4-fold for SUMO-tagged A'-A' with $l_3$ and 7.5-fold for A'-A' with $l_3$, when compared to cells incubated with A' monomers. Native-PAGE analysis following protein purification (see "Methods" and Supplementary Fig. S9) revealed the presence of a distinct protein band, indicating that A'-A' dipeptide does not aggregate. To verify CC ligand-receptor binding, Cy3 (sulfo-Cyanine3 NHS ester)-labeled SUMO-tagged A'-A' dipeptide (see "Methods" and Supplementary Fig. S10) was incubated with HEK293T cells expressing A-type receptors with $\alpha_2$ for 24 hours (Fig. 3d). Confocal fluorescence microscopy revealed ligand-receptor co-localization, indirectly confirming the presence of synthetic receptors in the cell membrane and validating the ligand-receptor interaction (Fig. 3e). On the contrary, cells expressing B- and Γ-type$_{JAK/STAT}$ receptors incubated with Cy3-labeled SUMO-tagged A'-A' dipeptide, showed no ligand-receptor co-localization (Supplementary Fig. S11). To summarize, these results collectively show that recombinantly expressed soluble, ditopic CC ligands can orthogonally bind to cognate CC-GEMS receptors, inducing dimerization and subsequent activation.

## Rerouting CC-GEMS signaling through alternative PLCG pathway activation

Next, we engineered CC-GEMS receptors to signal through the alternative PLCG pathway, in a manner similar to original GEMS activation[10]. In detail, the A CC-functionalised EpoR transmembrane scaffold of the GEMS receptor was fused to VEGFR2 to create A-type$_{PLCG}$ receptors (Fig. 4a and Supplementary Fig. S12). A SEAP reporter gene, engineered to be responsive to PLCG activation, through an NFAT (nuclear factor of activated T cells)-responsive minimal promoter ("Methods" and Supplementary Table S3) was used to prove A-type$_{PLCG}$ receptor activation, following incubation with A'-A' dipeptide ligand. We show a 7.6-fold increase in SEAP expression for HEK293T cells that have been transfected with A-type$_{PLCG}$ receptors and SEAP reporter gene and incubated with 0.12 $\mu$M of SUMO-tagged A'-A' dipeptide, compared to cells expressing A-type$_{PLCG}$ receptors and reporter gene that were not treated with ligand (Fig. 4b). Similarly, titration of SUMO-tagged A'-A' dipeptide in mammalian cells expressing A-type$_{PLCG}$ receptors and reporter resulted in robust receptor activation (see Supplementary Fig. S12). These findings suggest that CC-GEMS represent a versatile and modular platform, capable of interacting with various pathways to confer receptor activation.

## Scalability, orthogonality, and two-input logic using CC-GEMS

By exploiting the demonstrated ability of designated CCs to specifically and exclusively interact with a cognate partner[43,44,57,62], we next aimed to explore CC-GEMS receptor activation by an assortment of orthogonal dipeptide ligands, thereby showing the scalability and programmability of the platform. To this end, CC-GEMS receptors with linker $\alpha_2$, harboring A, B, or Γ CC domains were engineered (Supplementary Tables S1 and S2 and Supplementary Fig. S1) as well as the cognate, recombinant A'-A', B'-B', and Γ'-Γ' dipeptide ligands (see Supplementary Fig. S13 and inset in Fig. 1a for cognate pairs). To span individual CCs, we opted for the semi-rigid, helical linker $l_2$ (Supplementary Table S7), as previous research has shown increased expression of proteins containing helical linker domains in mammalian cells[63-65]. To assess CC-GEMS receptor activation from the designed cognate ligands, HEK293T cells were transfected with A-type$_{JAK/STAT}$, B-type$_{JAK/STAT}$, or Γ-type$_{JAK/STAT}$ receptors, STAT3 and SEAP reporter gene and were subsequently incubated with A'-A', B'-B', or Γ'-Γ'

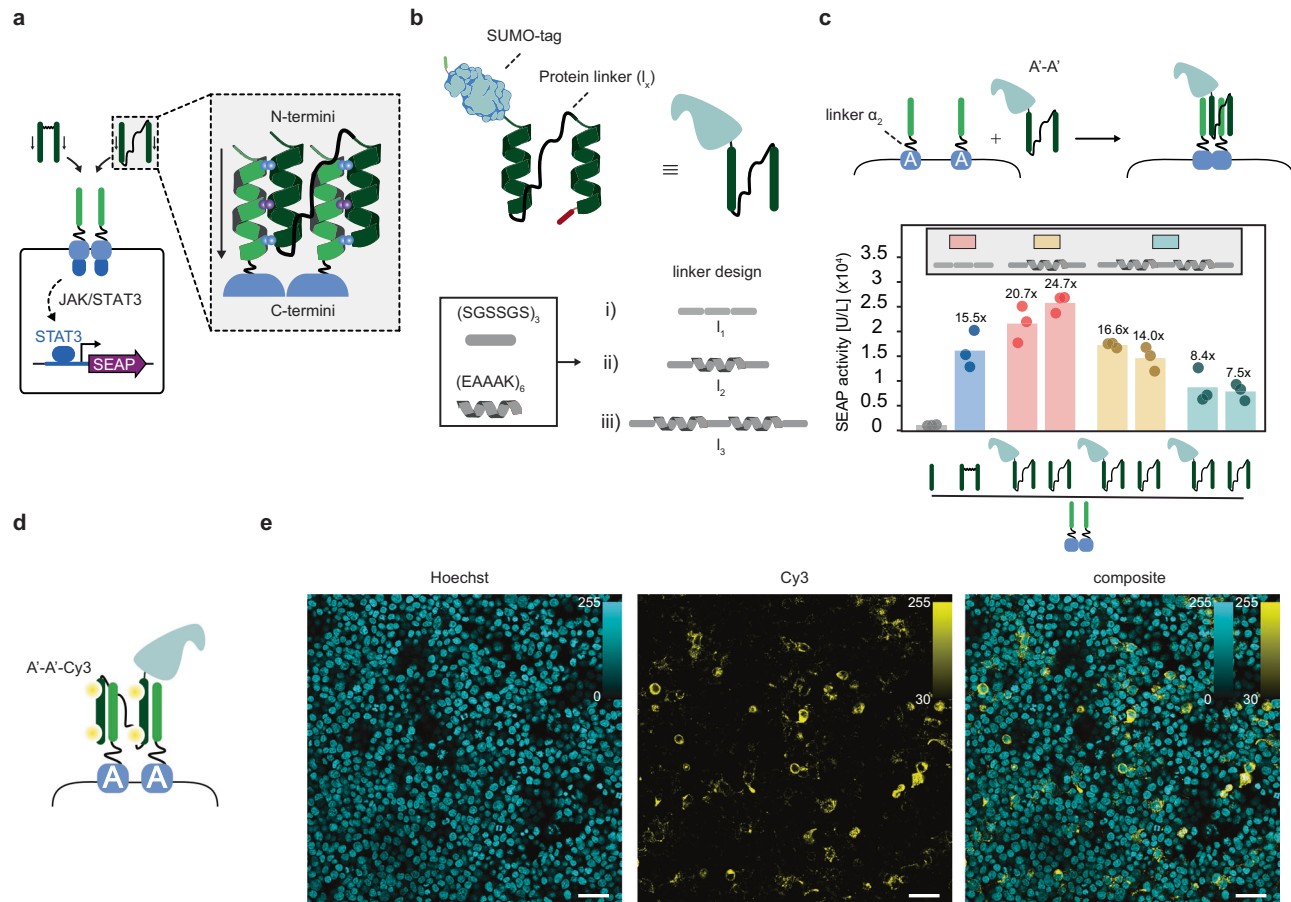

**Fig. 3 | Ditopic CC ligands with various protein linkers activate cognate receptors. a** Alternative ditopic CC ligands expressed in bacteria are engineered by fusing the N-terminus of a monomeric CC to the C-terminus of another, using various protein linkers. Arrows indicate N- to C- terminus directionality. **b** Ditopic ligands were expressed in *E. coli*, after SUMO-tagging. Flexible (SGSSGS) and more rigid (EAAAK) sub-units[58,59] were used to engineer a total of three linkers ($l_x$) (see "Methods" and Supplementary Table S7) **c** SEAP activity [U/L] in HEK293T cells transiently transfected with A-type$_{JAK/STAT}$ receptor with linker $\alpha_2$ (8 aa, GS repeats). Following transfection, cells were incubated with either 0.12 μM purified A'-A' dipeptide ligand, with or without a SUMO tag with various linker lengths ($l_1$: pink bars, $l_2$: yellow bars, or $l_3$: light blue bars), 0.12 μM A'-A' reference ligand (ref.: blue bar; see Fig. 2c), or 0.24 μM A' monomeric CC (see "Methods"). SEAP expression was monitored to assess receptor activation. Bars indicate mean activity; individual

data points represent independent triplicates, performed on the same day. Fold changes are calculated against cells incubated with A' monomeric CC and are shown above bars (for statistical analysis, see Supplementary Table S4).
**d** HEK293T cells expressing A-type$_{JAK/STAT}$ receptor with $\alpha_2$ were incubated with Cy3-labeled SUMO-tagged A'-A' dipeptide with $l_2$ (see "Methods") and imaged with a confocal fluorescent microscope. **e** Fluorescence confocal micrograph images of HEK293T cells expressing A-type$_{JAK/STAT}$ receptor with $\alpha_2$ following incubation with Cy3-labeled SUMO-tagged A'-A' dipeptide with $l_2$ ($n = 1$, see "Methods"). Cells were stained with Hoechst stain (blue). Cy3 in yellow. Scale bar (50 μm) is shown on the bottom-right of the image and intensity bar on top-right. Cy3 excitation: 553 nm, emission: 570–620 nm. Hoechst excitation: 405 nm, emission: 410–450 nm. Source data are provided as a Source Data file.

dipeptide ligands ("Methods"). Co-localization of ligand and receptor was verified for cognate pairs by fluorescence confocal microscopy of cells expressing the receptor incubated with fluorescently labeled ligands (Fig. 3e, Supplementary Figs. S11, S14, and S15, and "Methods"). SEAP quantification in cell medium showed that only cognate dipeptide ligands can activate the complementary CC-GEMS receptors, while non-cognate pairs result in negligible activation (see Fig. 5a and Supplementary Table S4). Interestingly, A'-A' dipeptide-A-type$_{JAK/STAT}$ receptor pairing resulted in the highest activation, followed by Γ'-Γ' dipeptide-Γ-type$_{JAK/STAT}$ receptor pairing and B'-B' dipeptide-B-type$_{JAK/STAT}$ receptor pairing; approximately 30-fold, 16-fold, and fivefold increase respectively. The higher binding affinity of A':A when compared to B':B and Γ':Γ has been previously demonstrated[44], suggesting that the increase in receptor activation could be correlated to the binding affinity of individual CCs. In addition, incubating A-type$_{JAK/STAT}$ receptor-expressing cells with an equimolar mixture of A'-A', B'-B' and Γ'-Γ' ligands exhibited similar activation levels to incubation with A'-A' ligand alone (Supplementary Fig. S16); demonstrating that non-

cognate CC dipeptides do not inhibit cognate dipeptide receptor binding. Collectively these data establish CC-GEMS as a scalable, tunable, and orthogonal platform for synthetic receptor activation that could be exploited to engineer intercellular communication between mammalian cells based on diffusible ligands (*vide infra*).

The orthogonality of cognate CCs[43,44,57,62] makes CC-GEMS an ideal platform for the implementation of Boolean logic. For instance, engaging an assortment of designed CC-GEMS receptors and dipeptides can enable receptor activation based on combinatorial ligand sensing, and OR gate logic operations (Fig. 5b). To show proof of the principle of the capacity of CC-GEMS to compute OR gate logic, we expressed two ditopic, SUMO fused B'-A' and A'-Γ' ligands (Supplementary Fig. S17a, S17b), able to dimerize cognate B-type$_{JAK/STAT}$ and A-type$_{JAK/STAT}$ as well as A-type$_{JAK/STAT}$ and Γ-type$_{JAK/STAT}$ receptors respectively (Supplementary Fig. S17c). We then transfected HEK293T cells with A-type$_{JAK/STAT}$, B-type$_{JAK/STAT}$, and Γ-type$_{JAK/STAT}$ receptors, *STAT3* and *SEAP* reporter gene (see "Methods") and subsequently incubated them with either B'-A' or A'-Γ' ligands with a SUMO

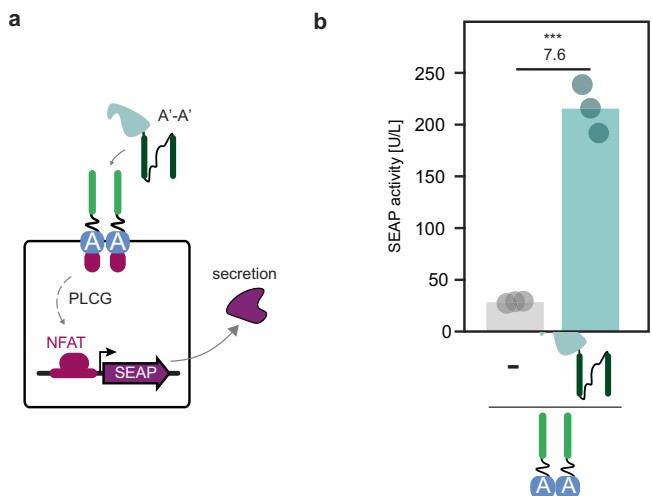

**Fig. 4 | CC-GEMS signaling can be rerouted through the alternative PLCG pathway. a** Schematic overview of the reporter system to monitor CC-GEMS receptor activation using phospholipase C gamma (PLCG) intracellular signaling. HEK293T cells were transfected with A-type$_{PLCG}$ receptors with linker $\alpha_2$ and a *SEAP* reporter gene, engineered to be responsive to PLCG activation, through an NFAT (nuclear factor of activated T cells)-responsive minimal promoter ("Methods"). Cells were subsequently treated with 0.12 µM SUMO-tagged A'-A' with I$_2$. **b** SEAP activity [U/L], measured 48 hours following transfection (see "Methods"). Fold change and significance (two-tailed, unpaired *t* test, *P* = 0.0002) is noted above bars. ns: *P* > 0.05, *P* ≤ 0.05, **P* ≤ 0.01, ***P* ≤ 0.001 (see Supplementary Table S6). Bars indicate mean activity; individual data points represent independent triplicates, performed on the same day. Source data are provided as a Source Data file.

tag, or an equimolar mix of both (Fig. 5c). When cells expressing A-, B-, and Γ-type$_{JAK/STAT}$ receptors were incubated with either 0.03 µM of SUMO fused B'-A' or A'-Γ' ligands or 0.015 µM of both ligands (for a total of 0.015 µM), receptor activation was observed (see Fig. 5c and Supplementary Table S4). In addition, incubation with 0.5 µM of either B'-A' or A'-Γ' ligands with a SUMO tag resulted in an almost three-fold increase in receptor activation, while incubation with an equimolar mix of both ligands for a total of 0.5 µM resulted in a 10.8-fold increase (see Supplementary Fig. S18). To rationalize the lower receptor activation due to incubation with either B'-A' or A'-Γ' ligands in cells transiently expressing A-, B-, and Γ-type$_{JAK/STAT}$ receptors, we performed fluorescence confocal microscopy experiments by incubating these cells with fluorescently labeled ligands (see "Methods" and Supplementary Fig. S19). Our results reveal a heterogenous population consisting of cells expressing a single receptor or combinations of two or three receptors, in accordance with literature[66]. When cells expressing one type of receptor were compared to cells expressing three individual receptor types as it is the case for the OR gate, using confocal fluorescence microscopy, we noticed a marked decrease in individual receptor availability for cells expressing three receptor types (Supplementary Figs. S20–S22). Since receptor availability on the cell membrane is a limiting factor for receptor activation, future research could focus on unraveling the optimal receptor membrane density needed to achieve the desirable response.

Next, we established combinatorial CC ligand sensing by means of AND gate logic, by expressing A'-Γ and Γ'-A' dipeptide ligands as a fusion protein with SUMO (Supplementary Fig. S23). In this design, the Γ:Γ' pairing facilitates inter-ligand dimerization, resulting in a CC-CC complex with two accessible A' domains, available for receptor binding (see Fig. 5b, right panel). For this ligand design, we opted for a shorter linker I$_4$ spanning the individual CCs, assuming that the Γ:Γ' interaction can act as a natural spacer allowing parallel orientation for the two available A' CCs, rendering the use of a longer linker unnecessary (Supplementary Table S7). Using Native-PAGE we reveal that, following

incubation of purified, equimolar SUMO-tagged A'-Γ and Γ'-A' ligands for 1 hour, the A'-Γ:Γ'-A' complex is assembled (see Supplementary Fig. S24a, S24b). Size Exclusion Chromatography (SEC) analysis of incubated, equimolar concentrations of SUMO-tagged A'-Γ and A'-Γ' dipeptides shows that incubation of dipeptides overnight results in complex formation (see Supplementary Fig. S24c, S24d), confirming the SDS-PAGE results. As Fig. 5d illustrates, only in the presence of both SUMO-tagged A'-Γ and Γ'-A' ligands, receptor activation in HEK293T cells transiently expressing A-type$_{JAK/STAT}$ receptor, *STAT3* and *SEAP* reporter gene is observed. In detail, incubation with both A'-Γ and Γ'-A' resulted in an approximately 30- to 45-fold increase compared to incubation with A'-Γ or Γ'-A' alone or when no ligand was present. Additionally, we incubated HEK293T cells expressing A-type$_{JAK/STAT}$ receptor, *STAT3* and *SEAP* reporter gene with A'-Γ and Γ'-A' dipeptide ligands, where the SUMO tag was cleaved and observed robust reporter gene expression, with an approximate 45-55-fold receptor activation (Supplementary Fig. S25) suggesting that the presence of a SUMO tag does not inhibit receptor activation. Thus, our data shows that both Boolean OR and AND operations can be engineered at the receptor level utilizing the CC-GEMS platform.

## Establishing intercellular communication using CC-GEMS

The design of intercellular communication between mammalian cells has been limited by the availability of orthogonal signaling mediators. Therefore, we sought to realize a synthetic, intercellular, communication system in mammalian cells that consists of a sender population secreting soluble, bifunctional CC ligands and a receiver cell population expressing the cognate CC-GEMS receptor and reporter gene. To prove that CC dipeptide ligands secreted by mammalian cells can activate CC-GEMS receptors, we stably transfected HEK293T cells to express and subsequently excrete SUMO-tagged A'-A' dipeptide with linker I$_2$ (Senders; Supplementary Fig. S26). Following lentiviral transduction (see "Methods"), we obtained a heterogenous sender population, with 84.4% of cells expressing SUMO-tagged A'-A' ligand— as quantified by the detection of EmGFP expressed as a bicistron (see Supplementary Fig. S27a, S27b). The SUMO-tagged A'-A' dipeptide was successfully recovered from the medium of sender cells following purification (Supplementary Fig. S27c). Next, the SUMO-tagged A'-A' purified ligand was added to the medium of HEK293T cells transiently transfected to express A-type$_{JAK/STAT}$ receptor with $\alpha_2$, *STAT3* and *SEAP* reporter gene. Our data shows that A'-A' dipeptide ligands expressed in mammalian cells can activate A-type$_{JAK/STAT}$ receptors, albeit in a moderately decreased fashion compared to A'-A' dipeptide ligands expressed in bacteria (Supplementary Fig. S27d). To achieve transcriptional control over CC ligand expression, we placed the A'-A' gene under the control of a doxycycline-induced cytomegalovirus (CMV) promoter[67,68] ("Methods" and Supplementary Fig. S26) and transduced HEK293S GnTi⁻ TetR cells[67] ("Methods" and Supplementary Fig. S28a, S28c). HEK293S GnTi⁻ TetR cells were stably transfected with the pcDNA6/TR vector, resulting in constitutive expression of TetR that bind to the tetracycline response element (TRE) downstream of the CMV promoter, repressing transgene transcription[69]. Next, cells were sorted as single cells to obtain monoclonal cell lines, of which a monoclonal population with minimal leakage for expression of the transgene in the absence of doxycycline was chosen (Supplementary Fig. S28b), and doxycycline-induced expression was assessed (Supplementary Fig. S28d, S28e). The selected population attained 99.4% ligand expression upon induction with doxycycline, as quantified by the detection of EmGFP expressed as a bicistron. Receiver cells were constructed as previously, by transiently transfecting A-type$_{JAK/STAT}$ receptor, *STAT3*, and *SEAP* reporter genes (see "Methods").

To demonstrate the ability of the CC-GEMS platform to facilitate intercellular communication, we co-cultured sender and receiver cells (see "Methods") in the presence and absence of inducer (Fig. 6a) and quantified the expression of the reporter gene in the cell culture

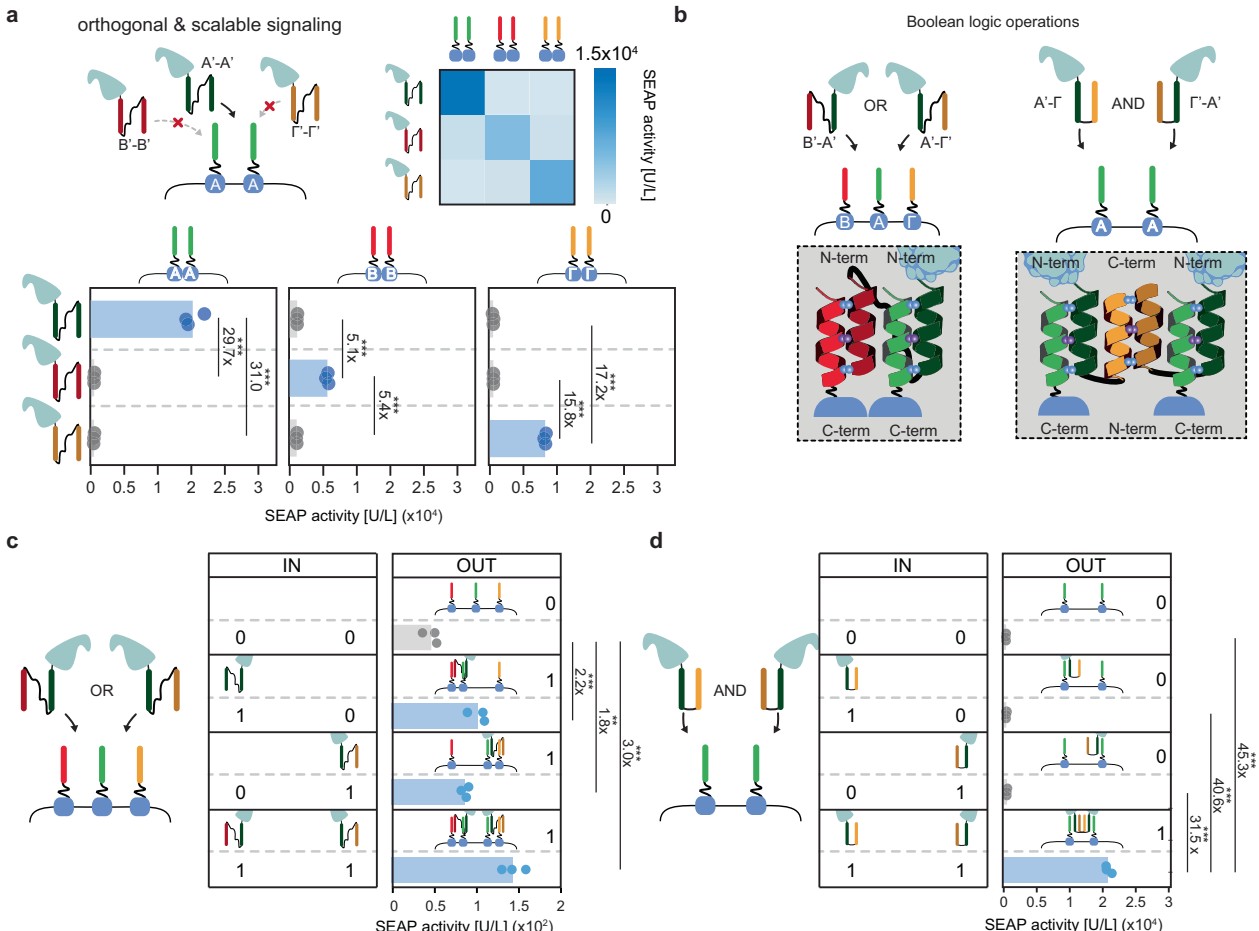

**Fig. 5 | The CC-GEMS platform is scalable and programmable. a** SEAP activity [U/L] in HEK293T cells transiently transfected with CC-GEMS receptors (bottom panel; A-type$_{JAK/STAT}$: left, B-type$_{JAK/STAT}$: middle, Γ-type$_{JAK/STAT}$: right) with linker $\alpha_2$ (8 aa, GS repeats), and subsequently incubated with 0.12 μM purified A'-A', B'-B', or Γ'-Γ' SUMO-tagged dipeptide ligand, with $l_2$ linker (see "Methods"). The heat-map on the upper right corner depicts mean SEAP activity per receptor–ligand pair (white: low SEAP activity, blue: high SEAP activity). **b** Schematic representation of envisioned Boolean logic operations. For the OR gate, HEK293T cells are transiently transfected with A-, B- and Γ-type$_{JAK/STAT}$ receptors followed by addition of ligands B'-A' and/or A'-Γ' to cell culture. For the AND gate, HEK293T cells are transiently transfected with A-type$_{JAK/STAT}$ receptor and later incubated with A'-Γ and A'-Γ' ligands. **c** SEAP activity [U/L] in HEK293T cells transiently transfected with A-, B- and Γ-type$_{JAK/STAT}$ receptors with linker $\alpha_2$ (8 aa, GS repeats), and afterwards incubated

with 0.03 μM SUMO-tagged purified ligand A'-B' or A'-Γ', or 0.015 μM A'-B' and 0.015 μM A'-Γ' ligands with linker $l_2$ (see "Methods"). Left panel (IN; input) denotes the presence (1) or absence (0) of ditopic ligand. Right panel (OUT; output) shows SEAP activity [U/L]. **d** SEAP activity [U/L] in HEK293T cells transiently transfected with A-type$_{JAK/STAT}$ receptors with linker $\alpha_2$ (8 aa, GS repeats), and incubated with 0.12 μM SUMO-tagged purified ligand A'-Γ and/or Γ'-A' with linker length $l_4$. Left panel (IN; input) denotes the presence (1) or absence (0) of ditopic ligand. Right panel (OUT; output) shows SEAP activity [U/L]. Bars indicate mean activity; individual data points represent independent triplicates, performed on the same day. Fold change and significance (one-way ANOVA with Dunnett's or Tukey's multiple comparison test) is noted besides the bars. ns: $P > 0.05$, *$P \leq 0.05$, **$P \leq 0.01$, ***$P \leq 0.001$. (see Supplementary Table S4). Source data are provided as a Source Data file.

medium. Our data show that intercellular communication can be achieved by co-culturing sender and receiver cells and that reporter gene expression can be temporally controlled with an ON/OFF switch (Fig. 6b). In detail, we observe a 6.6-fold increase in SEAP activity for receiver cells cultured with senders in the presence of dox compared to a co-culture with senders in the absence of the inducer and 9.7-fold increase compared to receiver cells cultured with control 293S GnTi⁻ cells; a response that is similar to previously engineered paracrine synthetic communication systems[31,33,34,36,37,42], ranging from 1.5- to 20-fold in performance. To understand the origin of this activation level, we plated $4.5 \times 10^5$ sender cells incubated with doxycycline and measured 0.14 μM of secreted SUMO-tagged A'-A' dipeptide after 48 h, by means of western blotting using an anti-Smt3 antibody ("Methods" and Supplementary Fig. S29). Although this concentration should result in optimal receptor activation, we notice a 14.8-fold decrease compared to receptor activation upon external addition of 0.12 μM of ligand obtained from expression in bacteria (Supplementary Fig. S30). We hypothesize that this lower receptor activation compared to the

external addition of ligand is due to the slow sustained secretion of the ligand from senders that progressively activates the receptors.

Next, we aimed to demonstrate that CC-GEMS is a suitable platform for engineering co-cultures that can perform distributed Boolean logic. As a proof of concept, we engineered a minimal three-cell population system that consists of two senders and a receiver cell population that together can execute AND gate logic (Fig. 6c). In detail, sender population 1 was created to secrete SUMO-tagged A'-Γ ligand with $l_4$ (stable transfection; Methods, Supplementary Figs. S31 and S32) and sender population 2 was engineered to secrete SUMO-tagged A'-Γ' ligand with $l_4$ ("Methods" and Supplementary Figs. S31 and S33). As depicted in Fig. 6d, receiver cells incubated with both sender populations (senders 1 and 2) show a significant, 7- to 9.6-fold increase, in reporter gene expression compared to receiver cells incubated with either sender population 1 or sender population 2 and control HEK293S GnTi⁻ cells. Quantification of the SUMO tag in the medium ("Methods" and Supplementary Fig. S34) showed the presence of 0.12 μM SUMO-tagged ligands. Considering that each

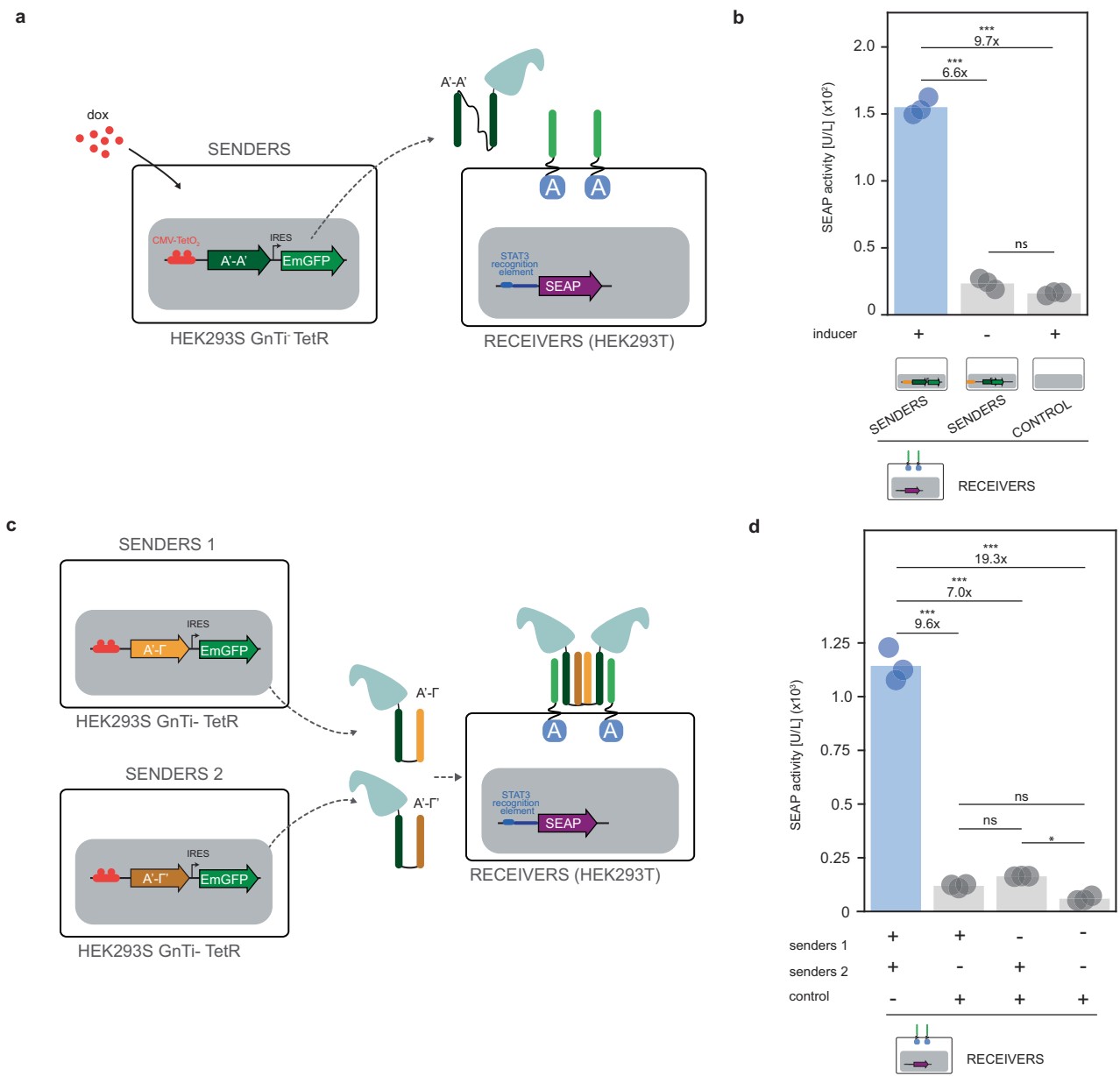

**Fig. 6 | Engineering intercellular communication using CC-GEMS. a** Intercellular communication is achieved by engineering sender HEK293S GnTi⁻ TetR cells that, upon induction with doxycycline (dox), secrete SUMO-tagged ditopic A'-A' ligand (see "Methods" and Supplementary Figs. S26 and S28) and receiver HEK293T cells expressing the receptor, *STAT3* and reporter gene *SEAP* (see Methods). **b** SEAP activity [U/L] in co-cultures of $1.5 \times 10^5$ receiver and $6 \times 10^5$ sender cells or control (untransduced HEK293S GnTi⁻ TetR) following incubation with and without dox (see "Methods"). **c** AND gate logic is achieved by engineering sender 1 (see Supplementary Fig. S32) and sender 2 (see Supplementary Fig. S33) HEK293S GnTi⁻

cells that secrete SUMO-tagged ditopic A'-Γ and Γ'-A' ligands with l₄ respectively (see "Methods") and receiver HEK293T cells. **d** SEAP activity (in U/L) for AND gate logic, after incubation of $1.5 \times 10^5$ receivers with $5 \times 10^5$ senders 1, $5 \times 10^5$ senders 2 and/or control, (untransduced HEK293S GnTi⁻ cells) for 48 hours (see "Methods"). Bars indicate mean activity; individual data points represent independent triplicates, performed on the same day. Significance (one-way ANOVA with Tukey's multiple comparison test) and fold change is noted above the bars. ns: $P > 0.05$, *$P \leq 0.05$, **$P \leq 0.01$, ***$P \leq 0.001$ (see Supplementary Table S4). Source data are provided as a Source Data file.

complex possesses two SUMO-tags, we expect the approximate final concentration of the complex to be near 0.06 µM. Collectively our results establish CC-GEMS as a scalable platform for engineering intercellular communication at the receptor level based on soluble, orthogonal ligands that can perform distributed Boolean operations.

## CC-GEMS can control the secretion of therapeutic ligands

To demonstrate the potential of CC-GEMS to be used for therapeutic purposes, we re-engineered the system to secrete a therapeutic protein as a response to a cognate ditopic CC ligand (Fig. 7a). In detail,

HEK293T cells were transiently transfected to express A-type_{JAK/STAT} receptor, *STAT3* and interleukin-10 (IL-10) under the control of a *STAT3* recognition element (Supplementary Fig. S35 and "Methods") and were subsequently treated with cognate A'-A' SUMO-tagged and non-cognate SUMO-tagged Γ'-Γ' with linker l₂ ditopic ligands. Our data shows that only upon addition of cognate ligand, is IL-10 secreted with a 15.5-fold increase compared to addition of non-cognate ligand and 17-fold increase compared to the negative control where no ligand was added (see Fig. 7b and Supplementary Table S4). These results demonstrate that CC-GEMS can secrete therapeutic proteins upon addition of cognate ligands.

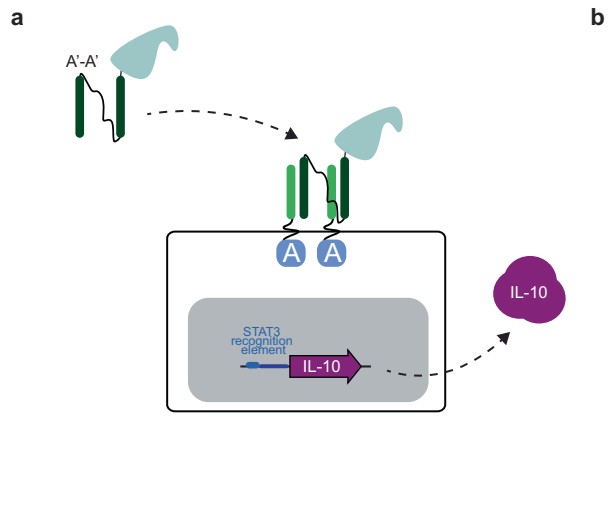
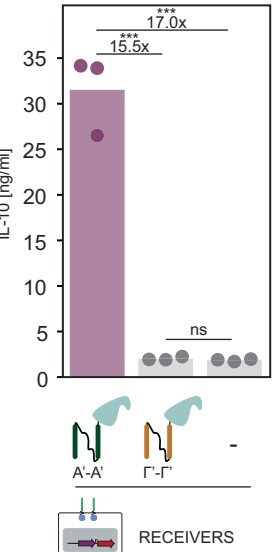

**Fig. 7 | CC-GEMS as a platform for therapeutic ligand production. a** Schematic representation of CC-GEMS platform utilized for IL-10 secretion. HEK293T cells are transiently transfected to express A-type$_{JAK/STAT}$ receptors with $\alpha_2$, *STAT3* and IL-10 under the control of *STAT3* recognition element (see Supplementary Fig. S35 and "Methods"). IL-10 is secreted upon the addition of SUMO-tagged A'-A' dipeptide with l$_2$. **b** IL-10 quantification measured by ELISA (see "Methods") in HEK293T cells transiently transfected with A-type$_{JAK/STAT}$ receptors with linker $\alpha_2$ (8 aa, GS repeats), and afterwards incubated with 0.12 µM purified SUMO-tagged ligand A'-A' or Γ'-Γ' with linker l$_2$ (see Supplementary Fig. S5b) or no ligand ("Methods"). IL-10 calibration curve is shown in Supplementary Fig. S36. Individual data points represent independent triplicates, performed on the same day. Fold change and significance (one-way ANOVA with Tukey's multiple comparison test) is noted above bars. ns: $P > 0.05$, *$P \leq 0.05$, **$P \leq 0.01$, ***$P \leq 0.001$ (see Supplementary Table S4). Source data are provided as a Source Data file.

## Discussion

Engineering synthetic communication in mammalian cells has the potential to uncover the fundamental principles of mammalian cell–cell communication circuits and provide a basis for the engineering of specialized cell communication networks with custom-defined functionalities. Here, we developed CC-GEMS, a modular mammalian synthetic communication platform based on designed, tunable, orthogonal, and diffusible ligands that can perform distributed Boolean logic operations at the receptor level. Our strategy is based on the functionalization of synthetic GEMS receptors[10] with CCs that are engineered to orthogonally bind to a selected partner[43,44]. The inherent orthogonality of CC pairing potentiates the engineering of a cell network that allows for targeted and specific communication between cells. In addition, the programmable design of large sets of CC heterodimers[45] enables the engineering of scalable cell circuit architectures. We demonstrate here that GEMS receptors functionalised with heterodimeric, cognate CC pairs result in robust receptor activation with little influence of the linker spanning the transmembrane domain of the receptor and the CC domain (Fig. 1d). The design of a CC dipeptide ligand that can be expressed allows for subsequent receptor activation in an orthogonal manner (see Figs. 3 and 5a), enabling the implementation of Boolean operations, in the form of AND and OR gates (see Fig. 5b–d). In addition, we show that CC-GEMS can function utilizing two distinct intracellular pathways, namely JAK/STAT and PLCG (Fig. 4). By engineering a sender–receiver population system (Fig. 6), where sender cells can secrete the ligand of choice under the control of a chemical switch, we provide evidence that our system can be used for the engineering of intercellular synthetic communication in mammalian cells. Finally, we show CC-GEMS-dependent therapeutic ligand expression, where CC-GEMS secretes IL-10 as a response to incubation with a cognate CC dipeptide (Fig. 7).

We envisage that CC-GEMS has several potential applications, including the design of cell-based therapeutic approaches. In detail, although autonomously regulated, population control in mammalian cells has been previously achieved, by repurposing the plant hormone

auxin[38], such an approach has limited scalability. We expect that quorum sensing in mammalian cells could be enabled by CC-GEMS and as such pave the way for the engineering of sophisticated systems of therapeutic cells that resemble the human immune system. While CC-mediated interactions have been recently used to engineer cellular assemblies in a helixCAM platform[70], CC-GEMS could be used to modulate cellular response depending on cell–cell interactions through membrane presented ditopic CC signaling mediators.

One of the prominent features of CCs is their designable orthogonality and tunability, which allows the selection of CCs with appropriate affinity[71] to obtain the desired response under the required conditions. In addition, introduction of combinations of several dipartite CCs and use of 4-helix bundles could enable combinations of more than two input signals and more versatile complex information processing. Theoretically, AND gate logic could be achieved using a set of original GEMS receptors responding to diverse ligands in a manner similar to the way CAR and SynNotch receptors were previously used[72,73], where one GEMS detects a cognate ligand and subsequently triggers the expression of a second GEMS that can be activated by a second ligand, producing the final output. However, such transcription-dependent logic introduces additional burden to the cell due to the introduction of two distinct receptor types and results in a delayed response. In contrast, AND gate logic in CC-GEMS occurs at the receptor level and can be engineered using a single CC-GEMS receptor type[46]. Furthermore, the ability of CC-GEMS to perform AND gate logic operations at the receptor level could allow the engineering of three-member cell populations that upon the sensing of two input signals result in a coordinated response.

Future research could focus on engineering alternative dimeric synthetic receptors to respond to CC modules. For instance, MESA (Modular Extracellular Sensor Architecture)[50] or DocTAR (Double-cut Transcription Activation Receptor)[74] receptors could similarly be re-engineered to express CCs as their extracellular domain that dimerize upon the addition of ditopic CC ligands. Such an approach has the potential to yield a maximally orthogonal communication system

based on synthetic receptors, since in the MESA and DocTAR architecture, signaling downstream from the engineered receptors is not subject to cross-talk with native cellular pathways. Additionally, engineering primary immune cells with CC-GEMS has the potential to show the ability of our platform to control responses in cell-based therapeutic systems. Since the immune compatibility of CCs has been previously shown[62], when CC protein-origamis were assessed in mice, we similarly expect no innate immune response upon introduction of CC-GEMS in vivo.

# Methods

## Chemicals and reagents
All reagents and solvents were obtained from commercial sources and were used without further purification.

## Gene construct design and preparation
Plasmids were constructed using the appropriate backbones for either transient or stable expression of the transgene (see Supplementary Table S3) and desired DNA (purchased from Integrated DNA Technologies or Genscript) with the use of restriction digestion, with standard restriction enzymes (New England BioLabs; HF enzymes were used when possible) and ligation with T4 DNA ligase (M0202S, New England BioLabs). All PCRs were performed with Q5 High-Fidelity DNA Polymerase (M0491S, New England BioLabs) or Phusion High-Fidelity DNA Polymerase (M0530S, New England BioLabs) according to the manufacturer's instructions. All plasmids used in this study are listed in Supplementary Table S8. Sequence verification of genes and constructs was done by Sanger sequencing (BaseClear).

## Cell culture and transient transfection
Human embryonic kidney (HEK293T) cells (validated and mycoplasm-free; ATCC® CRL3216™) were maintained in the growth medium, Dulbecco's Modified Eagle Medium (DMEM, 41966; Thermo Fisher Scientific) supplemented with 10% (v/v) Fetal Bovine Serum (FBS, S-FBS-SA-015; Serana) and 1% (v/v) antibiotic penicillin/streptomycin (pen/strep, 10,000 U/mL; 15-140-122, Gibco), under standard incubation conditions (37 °C, in a humidified atmosphere of 5% $CO_2$).

To prepare for transfection, cells were seeded at a density of $2.4 \times 10^5$ or $1.5 \times 10^6$ (for the experiments in Fig. 6) cells per well in a 24-well plate (Greiner) containing growth medium. The next day, cells were transfected with 500 ng plasmid DNA (193.8 ng per receptor dimer for a total of 387.6 ng, 96.1 ng *STAT3*-induced secreted alkaline phosphatase (*SEAP*) reporter plasmid pLS13 or pLS13-IL-10 and 16.3 ng *STAT3* expression vector pLS15; see Supplementary Tables S3 and S8) for 5 hours, using 1.25 µl of the transfection reagent lipofectamine 2000 (11668019, Life Technologies) in a total of 200 µl Opti-MEM™ I reduced serum medium (31985062, Thermo Fisher Scientific), according to the manufacturer's instructions. For experiments regarding transient transfection for receiver A-type$_{PLCG}$ cells, we transfected a total of 387.1 ng A-type$_{PLCG}$ receptor and 112.9 ng NFAT-induced secreted alkaline phosphatase (SEAP) reporter plasmid pHY30 (Supplementary Tables S3 and S8). Following transfection, cells were incubated with growth medium, containing the appropriate ligand for 48 hours; at 37 °C (Figs. 1, 2, 3c, 4, 6b, and 7; Supplementary Figs. S6, S7, S12, S27, and S30) or 30 °C (Figs. 3d, 5, and 6d; Supplementary Figs. S11, S14, S15, S16, S17, S18, S19, S20, S21, S22, and S25). We opted to perform these experiments at 30 °C, since the melting temperature ($T_m$) of the cognate B:B' and Γ':Γ' pairs has been reported to be around 40 °C[43]. Only in the case of the OR gate (Fig. 5c and Supplementary Figs. S17, S18, S19, S20, S21, and S22), a total of 693.8 ng DNA was transfected (193.8 ng per receptor dimer (3×) for a total of 518.4 ng, 96.1 ng *STAT3*-induced SEAP reporter plasmid pLS13 and 16.3 ng *STAT3* expression vector pLS15). When enhanced green fluorescent protein (EGFP) was used as a transfection control, $2.4 \times 10^5$ HEK293T cells seeded in a 24-well plate were transfected with 500 ng

of pHR-EGFPligand plasmid DNA (see Fig. 1 and Supplementary Fig. S3) as described above and transfection efficiency was assessed with flow cytometry. Experiments were undertaken by transiently transfecting individual clonal populations of cells, on the same day.

## SEAP quantification
SEAP concentration in cell culture medium was quantified in terms of absorbance increase due to para-nitrophenylphosphate (pNPP), using a commercially available SEAP assay kit (NBP2-25285, Novus Biologicals). Briefly, cell culture medium was harvested, and cell debris was removed by means of centrifugation. Subsequently, the medium was heat-inactivated at 65 °C for 30 minutes. To measure SEAP activity, cell culture medium was added to SEAP sample buffer and SEAP substrate in a transparent 96-well plate (Thermo Fisher Scientific), according to the manufacturer's instructions (in a final concentration of 0.83 mg/mL pNPP). Absorbance values were measured at 405 nm, at a controlled temperature of 25 °C, for 60 minutes, using a Tecan Spark 10 M plate reader (Tecan). To determine sample SEAP concentration, a calibration curve was constructed by titrating known concentrations of the hydrolyzed pNPP product para-nitrophenol (pNP) (Supplementary Fig. S2). Absorbance units are converted to amount of substrate conversion and SEAP activity is calculated from the slope of the time trace and expressed in units per litre (U/L). One unit is defined as the amount of enzyme that converts 1 µmole pNPP in 5 µL cell culture medium in 1 minute at 25 °C.

## IL-10 quantification
IL-10 concentration in cell culture medium was quantified in terms of absorbance increase (measured at 450 nm), using a commercially available ELISA kit (Human IL-10 ELISA MAX™ Deluxe Set, 430604, BioLegend). To determine IL-10 concentration in sample, a calibration curve was constructed (Supplementary Fig. S36) by titrating known concentrations of IL-10.

## Recombinant expression and purification of CC peptide ligands
A pET28a(+) vector (Supplementary Table S3) encoding the A' monomeric CC fused to SUMO tag (Supplementary Fig. S5 and Supplementary Table S1) was transformed into *E. coli* BL21(DE3) (69450, Novagen). A single colony of freshly transformed cells was cultured at 37 °C, in 500 mL 2× YT medium supplemented with 50 µg/mL kanamycin (CAS 25389-94-0, Merck). When the $OD_{600}$ (optical density measured at 600 nm) of the culture reached ~0.6, protein expression was induced by addition of β-D-1-thiogalactopyranoside (IPTG; 367-93-1, AppliChem), in a final concentration of 1 mM. The induced protein expression was carried out at 25 °C for ~18 hours and the cells were harvested by centrifugation at 10,000×*g* at 4 °C for 10 minutes. The pelleted cells were subsequently resuspended in BugBuster (5 mL/g pellet; 70584, Merck) supplemented with benzonase (5 µL/g pellet; Merck) and incubated for 15 minutes on a shaking plate. The suspension was centrifuged at 40,000×*g* at 4 °C for 30 minutes and the supernatant was subjected to $Ni^{2+}$ affinity chromatography on a gravity column. Briefly, after the supernatant was loaded on the column, the column was washed with washing buffer (1× PBS, 370 mM NaCl, 10% (v/v) glycerol, 20 mM imidazole, pH 7.4) and the protein was eluted with His-elution buffer (1× PBS, 370 mM NaCl, 10% (v/v) glycerol, 250 mM imidazole, pH 7.4). Cleavage of the N-terminal His-SUMO tag was performed by adding SUMO protease dtUD1 (1:500) to the elution fraction, while dialyzing (molecular weight cut-off (MWCO) 3.5 kDa, Thermo Fisher Scientific) against 2 L of dialysis buffer (50 mM Tris, 50 mM NaCl, pH 8.0) at room temperature for 16 hours. The concentrate was applied to a $Ni^{2+}$ column and the flow through, containing the CC ligand, was recovered, snap-frozen and stored at −80 °C for subsequent use. When CC monomers were used for synthesis of ditopic CC ligands, a final concentration of 2 mM TCEP (tris(2-carboxyethyl)phosphine)) reducing agent was

added to the protein aliquots. Protein concentration was calculated based on absorption at 280 nm in a Nanodrop 1000 spectrophotometer (ND-1000, Thermo Fisher Scientific), assuming an extinction coefficient calculated using the ProtParam tool on the ExPASy server. Protein purity was assessed on reducing SDS-PAGE.

Similarly, the various ditopic CC ligands with linker length $l_x$, were expressed in *E. coli* BL21(DE3) (Novagen). Various linkers $l_x$ were designed from a combination of $(EAAAK)_6$[58] and $(SGSSGS)_3$[59] monoblocks (see Supplementary Table S7). The calculated average distance of a single CC domain was approximated to be 4 nm, based on the 0.15 nm per residue length of an α-helix.

### Synthesis and purification of ditopic A'-A' CC ligands
Monomeric A' CC peptide, expressed in *E. coli* as described above, was buffered-exchanged to 100 mM Sodium Phosphate, 25 mM TCEP, pH 7 using repetitive washing and centrifugation with an Amicon 3 kDa MWCO centrifugal filter (UFC5003, Merck Millipore). For the synthesis of ditopic A'-A' CC ligand, 0.5 molar equivalent of homo-bifunctional bismaleimide linker (BS(PEG)$_3$; 21580, Thermo Fisher Scientific) was added to the protein solution and the reactants were incubated at 20 °C, 850 rpm for 3 h. Ditopic A'-A' CC ligands were purified using anion-exchange chromatography. Briefly, following equilibration of the anion-exchange column (0.5 mL strong anion-exchange spin column; 90010, Thermo Scientific) with purification buffer (50 mM Tris-HCl, pH 8.0), the protein mixture was loaded in 400 μL fractions, according to the manufacturer's instructions. Unreacted monomeric A' CCs were first eluted by treating the column with purification buffer supplemented with 200 mM NaCl and ditopic A'-A' CCs were eluted by treating the column with purification buffer supplemented with 300 mM NaCl. Purity of the recovered ditopic A'-A' CCs was assessed using SDS-PAGE under non-reducing conditions and purified products were snap-frozen and stored at −80 °C.

### Fluorescence confocal microscopy of cells expressing CC-GEMS receptors
To obtain the confocal microscopy images shown in Fig. 3 and Supplementary Figs. S11, S14, S15, S19, S20, S21, and S22, SUMO-tagged A'-A', B'-B', and Γ'-Γ' ligands (with $l_2$) were labeled using Cy3 (sulfo-Cyanine3 NHS ester; 11320, Lumiprobe), Cy5 (sulfo-Cyanine5 NHS ester; 13320, Lumiprobe), and AF-488 (AF-488 NHS ester; 11820, Lumiprobe) respectively (see Supplementary Fig. S10). Briefly, 30 μM ligand was incubated with 10× excess dye at 20 °C, for 2 hours (850 rpm), while protected from light. To remove excess dye, a Zeba™ spin desalting column (7 kDa MWCO) was used (89882, Thermo Fisher Scientific). HEK293T cells were transiently transfected with the corresponding receptors with $\alpha_2$ (as previously mentioned) in eight-well μ-slides (80807, ibidi), coated with Poly-L-Lysine solution (0.1% w/v; P8920, Merck). Forty-eight hours later, transfected cells were incubated with 0.12 μM labeled ligand, overnight, at 30 °C. The following day, cells were stained with 5 μg/ml Hoechst 33342 stain (5117, Biotechne) according to the manufacturer's instructions and imaged with the use of a Leica SP8 confocal microscope, using a Leica HC PL APO 20×/0.75 dry objective and 405 nm, 488 nm, 552 nm, and 638 nm continuous wave diode lasers. Prior to imaging, cells were fixed in 3.7% paraformaldehyde (PFA) solution. For image analysis, the Leica LAS X software (version 5.1.0) and ImageJ (1.53t; Java 1.8.0_322 (64-bit)) were used.

### Lentivirus production
Lentivirus was produced by co-transfecting HEK293T cells (ATCC® CRL3216™) that have reached ~80–90% confluency with the second generation pHR plasmid carrying the desired transgene (Supplementary Table S3) and the vectors encoding for the packaging proteins (pCMVR8.74; gift from Didier Trono (Addgene plasmid # 22036; http://n2t.net/addgene:22036; RRID: Addgene_22036) and VSV-G

envelope (pMD2.G, gift from Didier Trono (Addgene plasmid #12259; http://n2t.net/addgene:12259; RRID:Addgene_12259)) at a ratio of 1:2:1.2, using the FuGENE® HD transfection reagent (E2311, Promega) in Opti-MEM™ I reduced serum medium (31985062, Thermo Fisher Scientific). The following day, the medium was refreshed to DMEM (41966; Thermo Fisher Scientific) supplemented with 2% (v/v) FBS (S-FBS-SA-015; Serana) and cells were incubated under standard incubation conditions (37 °C, in a humidified atmosphere of 5% CO$_2$). Forty-eight hours later, the supernatant was harvested, and filtered with a 0.45-μm syringe filter (Merck). The produced lentivirus was pelleted by centrifugation at 50,000×*g*, 4 °C for 2 hours, resuspended in DMEM (41966; Thermo Fisher Scientific) supplemented with 2% (v/v) FBS (S-FBS-SA-015; Serana) and either used directly for cell line transduction or snap-frozen and stored at −80 °C.

### Sender cell line engineering
HEK293T cells (ATCC® CRL3216™) or human embryonic kidney 293 S lacking *N*-acetylglucosaminyltransferase I and expressing the Tet repressor protein (HEK293S GnTI⁻ TetR; obtained from the group of N. Callewaert at the VIB-UGent Center for Medical Biotechnology; nico.callewaert@ugent.vib.be)[75] were plated in 24-well plates (Greiner) and upon reaching 10-20% confluency were transduced with lentivirus harboring the SUMO-tagged A'-A' peptide with linker $l_2$ (Supplementary Fig. S26) or SUMO-tagged A'-Γ or Γ'-A' peptide with linker $l_4$ (Supplementary Fig. S31), under the control of the major immediate–early (MIE) human cytomegalovirus (CMV) enhancer and promoter and two *TetO* operator sequences[69] followed by an internal ribosome entry site (IRES) from encephalomyocarditis virus (EMCV) controlling the expression of an emerald green fluorescent protein (EmGFP). In total, 50–100 μL lentivirus was added to the cells, containing transduction medium (DMEM (41966; Thermo Fisher Scientific) supplemented with 10% (v/v) heat-inactivated FBS (S-FBS-SA-015; Serana)). Virus-containing transduction medium was replaced by growth media 48–72 hours post infection. Cells were bulk-sorted or sorted as single cells for EmGFP by fluorescence-activated cell sorting (FACS) in 1:1 conditioned to growth media (conditioned medium being medium that has been in contact with cells for 24 hours and subsequently harvested and filtered with a 0.45-μm syringe filter (Merck)). Cells were cultured under standard incubation conditions (37 °C, in a humidified atmosphere of 5% CO$_2$) and passaged upon reaching confluency (approximately every 2 days). Monoclonal cell lines were generated from the expansion of individually sorted cells that were left undisturbed in an incubator, under standard incubation conditions (37 °C, in a humidified atmosphere of 5% CO$_2$). HEK293S GnTI⁻ TetR cells were routinely cultured in growth medium with the addition of 1 μg/mL blasticidine S HCl (11583677, Gibco) for selection of the TetR trait.

### Mammalian protein expression and purification
For A'-A' CC protein expression and purification (see Supplementary Fig. S27), 3.72 x 10⁶ HEK293T A'-A' sender cells were seeded in a T175 cell culture flask (Greiner), containing 35 mL DMEM (41966; Thermo Fisher Scientific) supplemented with 2% (v/v) FBS (S-FBS-SA-015; Serana) and 1% (v/v) antibiotic penicillin/streptomycin (pen/strep, 10,000 U/mL; 15-140-122, Gibco). The cells were cultured at 37 °C, in a humidified atmosphere of 5% CO$_2$ for 4 days and the medium was harvested and cell debris was removed by centrifugation at 10,000×*g* for 10 min. Subsequently, medium was filtered with a 0.45-μm syringe filter (Merck) prior to loading to a Strep-tag affinity chromatography column. Gravity flow columns (2 mL) were prepared with 400 μL 50% Strep-Tactin® XT Superflow High-Capacity suspension (2-1208-002, IBA Lifesciences). Protein purification was performed according to the manufacturer's protocol. Washing was conducted with Buffer W (100 mM Tris, 150 mM NaCl, 1 mM EDTA, pH 8.0) and elution with Buffer BE (100 mM Tris, 150 mM NaCl, 1 mM EDTA, 50 mM D-biotin).

Purified protein was concentrated with the use of a 3 K Amicon® Ultra 0.5 mL Centrifugal filter (UFC5003, Merck), according to manufacturer's instructions. Proteins were buffered-exchanged in buffer containing 50 mM Tris, 50 mM NaCl, pH 8.0.

### SDS-PAGE and Native-PAGE

For SDS-PAGE analysis, 4−20% SDS-PAGE Mini-PROTEAN® TGX Precast gels (4561094, Bio-Rad) were used with running buffer (25 mM Tris, 192 mM glycine, 0.1% SDS, pH 8.3; Bio-Rad). Samples were added to SDS sample buffer (62.5 mM Tris-HCl, pH 6.8, 2% SDS, 25% (v/v) glycerol, 0.01% bromophenol blue, and 100 mM DTT) and denatured at 95 °C for 5 min. For Native-PAGE analysis, purified ligands were added to Native-PAGE sample buffer (75 mM Tris, 576 mM glycine, 30% (v/v) glycerol, 0.01% bromophenol blue) and run in a 4−20% SDS-PAGE Mini-PROTEAN® TGX Precast gels (4561094, Bio-Rad) with running buffer (25 mM Tris, 192 mM glycine). To assess A′-Γ:Γ′-A′ complex formation, equimolar concentrations of purified A′-Γ and Γ′-A′ ligands were first incubated at 30 °C for 1 hour and the formed complex was run in a Native-PAGE gel (Supplementary Fig. S24a). Protein visualization was done with Coomassie Brilliant Blue G-250 (1610406, Bio-Rad). Gels were analyzed with the ImageQuant 350 (GE Healthcare) or Amersham ImageQuant 800 (Cytiva). All procedures were carried out according to the manufacturer's protocols.

### Size-exclusion chromatography

Equimolar concentrations of SUMO-tagged A′-Γ and A′-Γ′ dipeptides with protein linker $l_4$ were incubated for 24 hours at 30 °C and were subsequently characterized using SEC (see Supplementary Fig. S24c). The analytical SEC data reported was performed on a NGC Chromatography System (Bio-Rad) using a Superdex 200 Increase 10/300 GL column (GE Healthcare). 1× PBS was used as the running buffer.

### Flow cytometry analysis and fluorescence-activated cell sorting

Cells were interrogated and sorted on a FacsAria III (BD Biosciences), operating at low-middle pressure, using a 70-μm nozzle. Cells were interrogated using a 488 nm laser and a 530/30 nm detector for EmGFP and a 561 nm laser and a 610/20 nm detector mCherry. A total of 10,000−100,000 events were recorded, from which 2D plots of the side-scattered light area (SSC-A) versus forward-scattered light area (FSC-A), as well as FSC-A versus forward-scattered light height (FSC-H) were obtained. As denoted in the figure captions, a first gate on FSC-A x SSC-A was constructed to identify the clonal population and a second gate on FSC-A x FSC-H was constructed to discriminate single cells from duplets.

### Mammalian A′-A′ dipeptide and A′-Γ- A′-Γ′-complex quantification using western blot

To quantify the amount of mammalian SUMO-tagged A′-A′ CC protein, $4.5 \times 10^5$ sender cells were plated in 24-well plates (Greiner; Supplementary Fig. S29a) in growth medium, supplemented with dox and incubated the cells under standard incubation conditions (37 °C, in a humidified atmosphere of 5% $CO_2$). Medium was harvested for quantification 48 hours later. To quantify the amount of mammalian A′-Γ- A′-Γ′-complex secreted in the medium by HEK293S GnTi- TetR sender cells, we harvested medium from the corresponding sender–receiver experiments following 2 days incubation at 30 °C (in a humidified atmosphere of 5% $CO_2$). Protein was quantified in the medium by means of western blotting analysis, using a primary antibody against the SUMO tag (Anti-Smt3 rabbit antibody; ab14405, abcam) and an anti-rabbit IgG secondary antibody (HRP conjugate; #7074, Cell Signaling Technology). Briefly, an SDS-PAGE gel was run for the mammalian proteins as well as a titration of bacterial SUMO-tagged A′-A′ dipeptide (Supplementary Figs. S29 and S30) that was later transferred to a PVDF (polyvinylidene difluoride) membrane, using an iBlot™ 2 Gel Transfer -Dry Blotting system (IB21001, Thermo Fisher Scientific) with

iBlot™ 2 Transfer Stacks (IB24001, Thermo Fisher Scientific), according to the manufacturer's instructions. Following blocking with blocking solution (5% solution of nonfat milk powder in 5 g/100 mL TBS-T (0.1% Tween20 in 8 g NaCl, 0.2 g KCl, 3 g Tris Base, pH=7.4)), for 1 hour, the membrane was incubated with 1:1000 primary antibody (diluted in blocking buffer), overnight at 4 °C. The following day, the membrane was washed 3× using TBS-T buffer for 10 minutes at room temperature, and subsequently incubated with secondary antibody (diluted 1:5000 in blocking buffer) for 1 hour at room temperature. After washing 3× using TBS-T buffer for 10 min at room temperature, substrate solution (Thermo Scientific™ SuperSignal™ West Pico PLUS Chemiluminescent Substrate, Thermo Fisher Scientific) was added and chemiluminescence was detected on an ImageQuant 800 imager (Cytiva), according to standard procedures. Background-subtracted density of the SDS-PAGE gel protein bands was determined as a function of area under the curve of intensity profile plots, using the gel analysis plugin of ImageJ 1.53a. Optical densities (in arb. units) of protein A′-A′ CC ligand expressed in *E. coli* with known concentrations were measured and linearly fitted to construct a calibration curve (Supplementary Figs. S29b and S34b) that was later used to determine A′-A′ CC dipeptide and A′-Γ- A′-Γ′-complex concentration expressed by mammalian cells.

### Sender–receiver co-culture

A total of $1.5 \times 10^5$ HEK293T cells were seeded in a 24-well plate (Greiner) in growth medium. To engineer receivers, the cells were transfected the following day with A-type$_{JAK/STAT}$ receptor, *STAT3*, and *STAT3*-induced SEAP reporter plasmid, as described above. After transfection, $6 \times 10^5$ sender cells or control HEK293S GnTi⁻ were added and the co-culture was incubated for 48 hours in 1 mL growth medium with or without the addition of 1 μg/mL dox at 37 °C, under standard incubation conditions. For the AND gate logic experiments presented in Fig. 6d, $1.5 \times 10^5$ receiver cells plated the day before, were incubated with a total of $5 \times 10^5$ sender 1 and $5 \times 10^5$ sender 2 or control HEK293S GnTi⁻ in the presence of 0.6 mL growth medium with dox (1 μg/mL) for 48 hours, at 30 °C, under standard incubation conditions. To increase the surface area available to cells seeded, we utilized Millicell Cell Culture Inserts (3.0 μm pore, translucent PET membrane; PTSP24H48, Merck). In detail, we seeded $3 \times 10^5$ sender 1 and $3 \times 10^5$ sender 2 or control HEK293S GnTi⁻ on the well-plate surface, as well as $2 \times 10^5$ sender 1 and $2 \times 10^5$ sender 2 or control HEK293S GnTi⁻ on the insert. The cell culture medium was subsequently used to determine the concentration of SEAP as described above. Experiments were undertaken by co-culturing individual clonal populations of cells, on the same day.

### Data analysis and statistics

Data were plotted with the use of MATLAB Toolbox Release R2019a for Windows (The MathWorks, Inc., Natick, MA, USA) and show the mean as a bar diagram overlaid with a dot plot of individual data points (for $n = 3$ biologically independent samples). Statistical analysis was done with GraphPad Prism 9.4.1 for Windows (GraphPad Software, San Diego, California USA, www.graphpad.com). All analyses were executed using unpaired, two-sided t test in case of comparing two experimental groups or One-way Analysis of Variance (ANOVA) with Tukey's multiple comparisons test (when comparing three or more experimental groups), Šidák's test (for planned comparisons) or Dunnett's test (when comparing means to a control group). Only values of $P < 0.05$ were considered statistically significant. No statistical method was used to predetermine sample size. No data were excluded from the analyses and experiments were not randomized. The Investigators were not blinded to allocation during experiments and outcome assessment. Flow cytometry data were analyzed with the use of FlowJo™ v10.7.1 Software (BD Life Sciences) for Windows. For image analysis, the Leica LAS X software (version 5.1.0) and ImageJ (1.53t; Java 1.8.0_322 (64-bit)) were used.

**Reporting summary**

Further information on research design is available in the Nature Portfolio Reporting Summary linked to this article.

## Data availability

Datasets generated and/or analyzed during the current study have been deposited in the Zenodo database at https://zenodo.org/record/8055628 (https://doi.org/10.5281/zenodo.10006328). Source data are provided with this paper.

## Materials availability

Unique biological materials, such as plasmids and cell lines are available from the corresponding author upon request. The following plasmids: pET28a(+)-SUMO-A' (Addgene plasmid #209115), pET28a(+)-SUMO-A'-l₁-A' (Addgene plasmid #209116), pET28a(+)-SUMO-A'-l₂-A' (Addgene plasmid #209117), pET28a(+)-SUMO-A'-l₃-A' (Addgene plasmid #209118), pET28a(+)-SUMO-B'-l2-B' (Addgene plasmid #209119), pET28a(+)-SUMO-Γ'-l2-Γ' (Addgene plasmid #209120), pET28a(+)-SUMO-A'-l2-B' (Addgene plasmid #209121), pET28a(+)-SUMO-A'-l2-Γ' (Addgene plasmid #209122), pET28a(+)-SUMO-A'-l4-Γ (Addgene plasmid #209123), pET28a(+)-SUMO-A'-l4-Γ' (Addgene plasmid #209124), pLeo-A-GEMS$_{JAK/STAT}$ (Addgene plasmid #209109), pLeo-A-α₂-GEMS$_{JAK/STAT}$ (Addgene plasmid #209110), pLeo-A-α₃-GEMS$_{JAK/STAT}$ (Addgene plasmid #209111), pLeo-B-α₂-GEMS$_{JAK/STAT}$ (Addgene plasmid #209112), pLeo-Γ-α₂-GEMSJAK/STAT (Addgene plasmid #209113), and pLeo-A-α₂-GEMS$_{PLCG}$ (Addgene plasmid #209114), pHR-SUMO-A'-l2-A'-IRES-EmGFP (Addgene plasmid #209125), pHR-SUMO-A'-l4-Γ-IRES-EmGFP (Addgene plasmid #209126), pHR-SUMO-A'-l4-Γ'-IRES-EmGFP (Addgene plasmid #209127), pLS13-IL-10 (Addgene plasmid #209128) are available from Addgene (see Supplementary Table S8).

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

## Acknowledgements

The authors thank R. Driessen for useful discussions regarding stable transfections, Jesse Lentjes for her help with protein expression, and

Indra Van Zundert for her help with confocal microscopy experiments. The authors would like to thank Leo Scheller and Martin Fussenenger for providing the original GEMS and reporter plasmids and the TU-Eindhoven iGEM 2022 team for providing us with the PLS13-IL-10 plasmid. This work was supported by the European Research Council (ERC project no. 101000199 AMIGA, awarded to T.F.A.d.G.) grant, the European Research Council (ERC project no. 899259 MaCChines, awarded to R.J.) grant, and by grants from the Slovenian Research and Innovation Agency (P4-0176, J7-4493, J1-2481, awarded to R.J.).

## Author contributions

A.M.P., G.A.O.C., and B.L.N. designed the study, performed experiments, and analyzed the data. A.M.P. wrote the manuscript. A.M.P., B.L.N., and T.J.M. established cell lines and performed related experiments. B.W.A.B., G.A.O.C., and A.M.P. analyzed and plotted all SEAP quantification experiments. B.V.E. and M.T.H.B. assisted in cloning and cell experiments. A.d.D. performed experiments related to protein characterization. O.M.J.A.S. and C.C.V.B. established the lentiviral production and provided key insights with regard to stable transfections. M.M. and R.J. provided critical feedback on experiments and revised the manuscript. T.F.A.d.G. conceived, designed and supervised the study, analyzed the data and wrote the manuscript.

## Competing interests

The authors declare no competing interests.
