## [Peer Review File · Nature Communications]

Reviewers' Comments:

Reviewer #1:

Remarks to the Author:

In this manuscript by Pistikou et al. titled "Engineering a Scalable and Orthogonal Platform for Synthetic Communication in Mammalian Cells", the authors engineer a platform for synthetic intercellular communication between mammalian cells that uses diffusible coiled-coil peptides and their cognate surface receptors. After a series of experiments building and characterising the system for peptide signalling, the authors implement OR and AND gate logic by reorganising orthogonal receptor-peptide pairs. Finally, they implement a 2 input (2 types of sender cells) distributed AND gate using a population of 3 cell-types.

In this work, the authors start with the previously published GEMS sensor platform that enabled sensing of several input molecules by surface receptors, followed by receptor dimerization and subsequent transgene expression in the sensor cells. Next, they change the surface receptors in the GEMS system by peptides from the NICP set. Through some interesting protein engineering approaches, they manage to build orthogonal peptide ligand-receptor pairs (Fig 1-3). However, beyond this point the data become a little less convincing. For example, two ON states of OR logic gate in Fig.4c are hard to distinguish from the OFF state. Similarly, the expression difference between the ON and some OFF states in the 3-cell AND gate is barely 1.3-fold.

Despite the final results being less categorical than one would have hoped for, the goals of this work are very relevant and the experimental work is highly involved. The ms is also well written. That said, there are several mistakes/ typos in the text, and problems with missing technical details and analyses, that the authors should address before some results can be fairly assessed. I list below several of my concerns regarding the manuscript, which I hope the authors will be able to improve upon.

Main points:

(1) It's unclear to me why some cell lines are transfected transiently and others stably? The authors should comment on this choice used for different experiments, especially since they argue that heterogeneity among transiently transfected cells results in lower activation.

(2) p4: "receptor activation is not critically dependent on linker length between the CC and EpoR domain of the receptor" I am not sure this conclusion can be reached by analysing just 4 length variants for one receptor domain. The authors should qualify it.

(3) Fig. 1 and others: Please include all the un-normalised SEAP expression values as Supp data.

(4) In the legends of Fig 1 and other figures, what does "independent triplicates" mean? These are cells from different transfection experiments, or three clonal populations? Were the independent experiments done on the same day, or on different days. These experimental details must be provided.

(5) Fig 2c and other figures: uncropped versions of all gels and blots shown should be provided as Supp data.

(6) p6. "STAT3 and SEAP reporter genes and incubated the transfected cells ..." Is STAT3 a reporter gene in this case?

(7) p7: The authors should comment on why linked length did not affect activation in Fig 1, but it does in Fig 2.

(8) Fig2e/ Fig 4 a,c,d: What are these SEAP values normalised to? As a general rule, wherever presenting normalised data, please state what it is normalised to (the mean of values for A:A' receptor dimer expressing cells?) and present the un-normalised values in Supp Data. Also, describe how the SEAP activity normalisation was done in the Methods section. Currently, pointers to this only appear in figure legends.

(9) Fig 4c: Is the SEAP expression between states 00 and 10 (or 00 and 01) statistically significant? Please present the p-values from the appropriate statistical tests. This also raises the question of when the authors do perform statistical testing and when they avoid it. Please be consistent throughout.

(10) p.12: "In this ligand design, we opted for a shorter linker I4 spanning the individual CCs" The authors should explain the rationale for this choice, given that I2 was used earlier for the OR gate and esp since this is a new linker length untested in Fig 3c.

(11) Fig 5b,d: Why did the authors switch to using absolute units for SEAP activity, as opposed to normalised activities used up to Fig.4. How do these values compare to the previously used normalisation control?

(12) p15: "receiver cell line exhibits leaky expression of the reporter gene" What has changed about the STAT3-SEAP construction that increases leakiness compared to the previous designs?

(13) p19: "cells were transfected with 500 ng plasmid DNA ... for five hours" Is this correct?

(14) The authors should explain the rationale for gating for single cells (FSC-A x FSC-H) shown in Fig S3.

(15) Sequences of all new plasmids made in this study should be listed in the supplementary materials. Do the authors plan to deposit their plasmids/ cell lines in accessible repositories?

(16) All source/ raw data should be made available as Supp data with the ms (journal guidelines permitting), or made available through a public repository.

Other points/ suggestions:

(17) Fig.1: There are too many diff colour schemes being used to depict the receptors, which is confusing. I would suggest that CC halves of a pair (e.g. A and A') and well as their ligand/s should be coloured using shades of the same colour.

(18) Fig.2 : I think the point about CC directionality could be better made by including throughout an arrow in the CC symbol.

(19) Fig2e: The symbols below the plot are a bit confusing. It may help to draw a horizontal line to group the concentrations 0 to 1 uM, indicating they are for the ligand dimer while only 2uM is for the monomer.

(20) Fig 2d: To ease understanding, please re-align the symbols below the plot such that all ligands are in the top row and all receptors are in the bottom row.

(21) Fig 3c: As before, please re-align the symbols below the plot such that all ligands are in the top row and all receptors are in the bottom row. Also, to be less confusing please move the legend schematic, currently presented as plot inset, to the outside and put the labels I1, I2, I3 in the figure.

(22) p19: "and 30oC (Figure 4" Missing information

(23) At several places in the methods section, references to supplementary figures/ tables seem to be incorrect. Please cross-check and fix them. A few examples below:

p18: "appropriate backbones for either transient or stable expression of the transgene (see Table S2)"

p19: "by titrating known concentrations of the hydrolysed pNPP product para-nitrophenol (pNP) (Supplementary Figure S1)"

p20: "A pET28a(+) vector (Supplementary Table S2) encoding"

(24) p20: "stored at 80oC for subsequent use" Typo

Reviewer #2:

Remarks to the Author:

Review

In this work, Pistikou et al. have engineered a scalable and orthogonal platform for synthetic communication in mammalian cells by re-purposing orthogonal coiled-coil (CC) interacting peptides for the GEMS (Generalized Extracellular Molecule Sensor) platform. Additionally, the authors engineered secreted ligands and used them as Boolean logic gate operators for cell communication. The paper is well-written and the data are clear and detailed. However, I have several concerns, most importantly regarding novelty.

Major comments:

1- The novelty of this work is limited as the authors just combined two published platforms in a slightly modified way (CC interacting peptides from Chen et al, Nature, 2018: PMID: 30568301 and the GEMS platform from Scheller et al, Nat Chem Biol. 2018: PMID: 29686358). The main novelty here seems to be the engineering of the secreted ditopic ligands that stimulate their cognate receptors by optimizing different linkers along with SUMO tag. This could be considered an incremental improvement over prior work.

2- Several groups have developed more robust and clinically relevant systems combining orthogonal ligands with cognate receptors in immune cells. This work has not even been cited, which would have exposed the limited novelty. In principle, these prior systems can be used in a similar way as presented in this work because they are orthogonal, secreted, and even immune-compatible (Kalbasi et al., Nature, 2022: PMID: 35676488; Sockolosky et al., Science, 2018, PMID:29496879; Zhang et al., Sci Transl Med, 202 : PMID: 34936380).

3- The author stated that this platform can be utilized in some applications, but a compelling application is missing as is convincing proof, that the system is superior to prior work. The Boolean logic gate is not a compelling application and it can be achieved in many other ways using existing tools, even for cell communication. For example, Fink et al, Nat Chem Biol. 2019: PMID: 30531965 designed a full set of orthogonal Boolean logic gates based CC interacting peptides; Again, this manuscript lacks novelty.

4- In the GEMS publication, the authors used several receptors that activate STAT3, NF- κ B, NFAT, or MAPK pathways. Is there any reason why the authors focused on the STAT3 pathway. Is their system incompatible with the other signalling pathways?

5- In figure 4, the author showed bitopic activators that stimulate the receptors. What is the advantage of these ligands over Sun-Tag-mCherry and PSA (prostate-specific antigen) that are shown in the original GEMS platform? In both cases, these proteins can be secreted and can be used as described in Fig 5. Using nanobodies is also scalable, even more than CC interacting peptides.

6- If the goal is to engineer secreted ligands as presented in figures 4 and 5, what is the purpose of figure 2 where the authors artificially synthesized a ditopic ligand using BM(PEG)₃?

7- While receptor activation is orthogonal, the STAT3 pathway is still endogenous and it can be activated by multiple physiological clues. This is especially true when applying this system in a therapeutic context.

Minor comments:

- Why do the authors normalize SEAP levels? Absolute values are needed to judge the leakiness of the system.

-The system presented in figure 5b seems to be very leaky (SEAP units) and the performance is really poor (less than two fold change between the middle columns- with and without doxycycline). This is not a convincing performance.

- In figure 5d, the comparison should be between cells co-cultured in the presence or absence of doxycycline and not between different sender cell types.

- This reviewer does not understand why SEAP activity suddenly converted to substrate cleavage in 1c and why you need a time course measurement. In fact, this figure is included in figures 1d and 1a.

-Methods: All reagents used should be exactly cited along with the company name and catalog

number.

We would like to thank both reviewers for their useful suggestions and constructive criticism that we believe helped improve the performance of our system as well as strengthen the main conclusions drawn from the study.

REVIEWERS COMMENTS AND AUTHOR'S ANSWERS

Reviewer #1 (Remarks to the Author):

In this manuscript by Pistikou et al. titled "Engineering a Scalable and Orthogonal Platform for Synthetic Communication in Mammalian Cells", the authors engineer a platform for synthetic intercellular communication between mammalian cells that uses diffusible coiled-coil peptides and their cognate surface receptors. After a series of experiments building and characterising the system for peptide signalling, the authors implement OR and AND gate logic by reorganising orthogonal receptor-peptide pairs. Finally, they implement a 2 input (2 types of sender cells) distributed AND gate using a population of 3 cell-types.

In this work, the authors start with the previously published GEMS sensor platform that enabled sensing of several input molecules by surface receptors, followed by receptor dimerization and subsequent transgene expression in the sensor cells. Next, they change the surface receptors in the GEMS system by peptides from the NICP set. Through some interesting protein engineering approaches, they manage to build orthogonal peptide ligand-receptor pairs (Fig 1-3). However, beyond this point the data become a little less convincing. For example, two ON states of OR logic gate in Fig.4c are hard to distinguish from the OFF state. Similarly, the expression difference between the ON and some OFF states in the 3-cell AND gate is barely 1.3-fold.

Despite the final results being less categorical than one would have hoped for, the goals of this work are very relevant and the experimental work is highly involved. The ms is also well written. That said, there are several mistakes/ typos in the text, and problems with missing technical details and analyses, that the authors should address before some results can be fairly assessed. I list below several of my concerns regarding the manuscript, which I hope the authors will be able to improve upon.

Main points:

(1) It's unclear to me why some cell lines are transfected transiently and others stably? The authors should comment on this choice used for different experiments, especially since they argue that heterogeneity among transiently transfected cells results in lower activation.

Reply:

We thank the reviewer for bringing this point to our attention, as we think explaining the rationale of the decision to move from transiently transfected cells to creating stable cell lines should be stated in the main text. In detail, we chose to repeat the experiments of the intercellular cell communication (sender-receiver experiments in Figure 5) using transiently transfected receiver cells and stably engineered senders. This choice was made in order to overcome the problems we identified related to inducible reporter gene leakage in our previously engineered receiver cell lines. Following the reviewer's suggestions and in an attempt to be consistent over the choice of transfection in receiver cells, we replaced the results in Figure 5 with those coming from transiently transfected receiver cell experiments. However, we opted for continuing with the stable sender cell lines, as we consider the lentiviral transduction of mammalian cells a cost-effective approach for the production of soluble ligands (in support of this claim see Elegheert et al., 2018; *Nat Protoc*, 10.1038/s41596-018-0075-9). In conclusion, we believe our adapted approach resulted in improved performance with respect to the intercellular communication system that is primarily due to resolving reporter gene-leakage related issues. We have added this rationale to the main text.

(2) p4: "receptor activation is not critically dependent on linker length between the CC and EpoR domain of the receptor" I am not sure this conclusion can be reached by analysing just 4 length variants for one receptor domain. The authors should qualify it.

Reply:

We agree with the reviewer that the conclusion drawn from our data in Figure 1d relating to the effect of linker length in heterodimeric receptor activation should be adjusted to specifically referring to the four linker lengths used in the study. We now highlight this in the main text (paragraph #3 of subsection "*Design principles of a coiled-coil functionalised-GEMS synthetic communication platform*" of the *Results and Discussion* section, highlighted in yellow): "*For the four different linker lengths used in this study, our data reveals that receptor activation is not critically dependent on linker length...*". Although we believe we covered a wide range of linker lengths (namely: zero, four, eight, and 27 amino acids), we think it is possible that further lengthening the linker will have an effect on receptor

activation. Therefore, we included the following discussion point (continuing in paragraph #3 of subsection “*Design principles of a coiled-coil functionalised-GEMS synthetic communication platform*” of the *Results and Discussion* section, highlighted in yellow): “However, since linker-length dependence has been reported in other synthetic receptors (Daringer et al., 2014, *ACS Synth Biol*, 10.1021/sb400128g), further research is needed to investigate a broader range of linker lengths and its influence on CC-GEMS receptor activation.”.

(3) Fig. 1 and others: Please include all the un-normalised SEAP expression values as Supp data.

Reply:

In response to this remark, as well as comments #8, #11 from this reviewer, and minor comment #1 from reviewer #2, we opted for showing the non-normalised versions of the data.

(4) In the legends of Fig 1 and other figures, what does "independent triplicates" mean? These are cells from different transfection experiments, or three clonal populations? Were the independent experiments done on the same day, or on different days. These experimental details must be provided.

Reply:

We thank the reviewer for this remark. These details have been added to the *Methods* Section (highlighted in yellow), as a last sentence in the subsection “*Cell culture and transient transfection*”, where we specifically state: “*Experiments were undertaken by transiently transfecting individual clonal populations of cells, on the same day.*” Similarly, we supplemented the subsection “*Sender-receiver co-culture*” of the *Methods* with the following sentence: “*Experiments were undertaken by co-culturing individual clonal populations of cells, on the same day.*” (last sentence, highlighted in yellow).

(5) Fig 2c and other figures: uncropped versions of all gels and blots shown should be provided as Supp data.

Reply:

Please find all uncropped versions of gels, blots and grayscale, unprocessed confocal fluorescence images as Supplementary Raw Data (SS1-SS17) in the Supplementary Information document.

(6) p6. "STAT3 and SEAP reporter genes and incubated the transfected cells ..." Is STAT3 a reporter gene in this case?

Reply:

Receptor activation occurs via the JAK/STAT signalling pathway, since the transmembrane domain of CC-GEMS is fused to the JAK/STAT signaling domain of IL-6RB. Following receptor activation, JAKs phosphorylate STAT3 that in its turn functions as a transcription factor for the SEAP reporter gene. STAT3 is transiently transfected in the cells along with the reporter gene SEAP to assist in this process. Therefore, only SEAP (and not STAT3) is the reporter gene. This has been clarified in the revised main text and the text has been corrected from stating: "STAT3 and SEAP reporter genes" to "STAT3 and SEAP reporter gene" (singular; the word "gene" referring to SEAP only). Additionally, the following sentence has been added "*Following receptor activation, STAT3 is phosphorylated and becomes a transcription factor for the SEAP reporter gene.*", in the second paragraph of the *Results and Discussion* sub-section "*Design principles of a coiled-coil functionalised-GEMS synthetic communication platform*" (marked in yellow).

(7) p7: The authors should comment on why linker length did not affect activation in Fig 1, but it does in Fig 2.

Reply:

We thank the reviewer for this comment, as we believe that the discrepancy between linker-dependent receptor activation should be addressed. In response to this comment, we supplemented the main text with the following speculative explanation point: "*...endogenous receptor dimerization occurs without conformational restriction induced by membrane organization which could result in a different receptor activation mechanism compared to receptor activation on the membrane. As a result, diverse outcomes related to linker-length dependent activation can occur between cognate CC-GEMS receptor pairs that already dimerize in the secretory pathway compared to a CC-GEMS receptors that dimerize on the membrane upon the addition of a cognate ligand. Furthermore, a cognate CC-GEMS heterodimer, such as A'-type:A-type receptor pair, will result in a different receptor dimer proximity compared to a homodimeric CC-GEMS that is activated by an external ligand, such as A-type receptor pairs bound to A'-A' dipeptide. The difference in receptor dimer proximity between a receptor heterodimer and a ligand activated CC-GEMS should be considered when considering the diverse outcomes of linker length in receptor activation*" (end of paragraph #2 of subsection "*Design of soluble ditopic CC ligands to activate CC-GEMS receptors*" of the *Results and Discussion*, marked in yellow).

(8) Fig2e/ Fig 4 a,c,d: What are these SEAP values normalised to? As a general rule, wherever presenting normalised data, please state what it is normalised to (the mean of values for A:A' receptor dimer expressing cells?) and present the un-normalised values in Supp Data. Also, describe how the SEAP activity normalisation was done in the Methods section. Currently, pointers to this only appear in figure legends.

Reply:

As stated up above (main point (3) of the same reviewer), non-normalized values of data are presented in the paper in place of normalized values.

(9) Fig 4c: Is the SEAP expression between states 00 and 10 (or 00 and 01) statistically significant? Please present the p-values from the appropriate statistical tests. This also raises the question of when the authors do perform statistical testing and when they avoid it. Please be consistent throughout.

Reply:

We thank the reviewer for this valuable comment. In order to investigate ligand-dependent receptor activation for three receptors (OR gate) further, we repeated the experiment using different amounts of ligand (a total of 0.03 μ M in Figure 4c and 0.5 μ M in Supplementary Figure S17). In both cases, the activation with one or two ligands was significantly different compared to the state when no ligand is added. The statistical testing summary is now reported in the Figures and the details of the tests (including ANOVA F-values, ANOVA p-values, and adjusted p-values for multiple comparisons) can be found in Supplementary Table S4.

As stated in the main text (paragraph #2 of subsection “*Scalability, orthogonality, and two-input logic bio-computations using CC-GEMS*” of the *Results and Discussion*, marked in yellow): “*Our results reveal a heterogenous population consisting of cells expressing a single receptor or combinations of two or three receptors, in accordance with literature (Materna et al., 2005, FEMS Microbiol Lett, 10.1016/j.femsle.2004.11.035.)*” Using confocal fluorescence microscopy, we now show that transient transfection of mammalian cells with three unique receptors (in our case A-, B-, and Γ -type receptors) results in a heterogenous population of cells expressing only one, only two, or all three receptors. Please find supportive data for this claim in Supplementary Figure S18. Additionally, we show that in the case of the OR gate, receiver cells express significantly fewer individual receptors (Supplementary Figures S19-S21). We therefore reasoned that: “*Since receptor availability on the cell membrane most likely is a limiting factor for receptor activation, future research could focus on unravelling the optimal receptor membrane density needed to achieve the desirable response.*” (end of paragraph #2 of

subsection *“Scalability, orthogonality, and two-input logic bio-computations using CC-GEMS”* of the *Results and Discussion*, marked in yellow).

(10) p.12: "In this ligand design, we opted for a shorter linker I₄ spanning the individual CCs" The authors should explain the rationale for this choice, given that I₂ was used earlier for the OR gate and esp since this is a new linker length untested in Fig 3c.

Reply:

We explain this designer choice in paragraph #3 of subsection *“Scalability, orthogonality, and two-input logic bio-computations using CC-GEMS”* of the *Results and Discussion* (marked in yellow) with the following point: *“...we opted for a shorter linker I₄ spanning the individual CCs, assuming that the F:F' interaction can act as a natural spacer allowing parallel orientation for the two available A' CCs, rendering the use of a longer linker unnecessary”*.

(11) Fig 5b,d: Why did the authors switch to using absolute units for SEAP activity, as opposed to normalised activities used up to Fig.4. How do these values compare to the previously used normalisation control?

Reply:

As stated up above (main point (3) and (8) of the same reviewer), non-normalized values of data are now presented in the paper, in place of normalized values. We observed that although sender cells that are incubated with doxycycline for 48 hours secrete approximately 0.14 μM of SUMO-tagged A'-A' dipeptide (quantification shown in Supplementary Figure S28), which should be enough to optimally activate the receptor, there is a significant decrease in activation compared to ligand-induced receptor activation (data presented in Supplementary Figure S29). We explain this decrease by anticipating that sustained release of ligand from the senders progressively activates the receptors, which is not the case for the immediate receptor activation by ligand added to the culture medium. This point is discussed in paragraph #2 of the subsection *“Establishing intercellular communication using CC-GEMS”* of the *Results and Discussion* (in yellow), where we state: *“In detail, we observe a 3.5-fold increase in SEAP activity for receiver cells cultured with senders in the presence of dox compared to a co-culture with senders in the absence of the inducer. To understand the origin of this activation level, we plated 0.45×10^6 sender cells incubated with doxycycline and measured 0.14 μM of secreted SUMO-tagged A'-A' dipeptide after 48 hours, by means of western blotting using an anti-Smt3 antibody (Methods and Supplementary Figure S28). Although this concentration should result in optimal*

receptor activation, we notice a 9.1-fold decrease compared to receptor activation upon external addition of 0.12 μ M concentration of ligand obtained from expression in bacteria (Supplementary Figure S29). We hypothesize that this lower receptor activation compared to external addition of ligand is due to the slow sustained secretion of ligand from senders that progressively activates the receptors.” We would like to thank the reviewer for his valuable insights and critical points, since we believe that those prompted us to undertake further experiments to support our hypotheses and further investigate our claims.

(12) p15: "receiver cell line exhibits leaky expression of the reporter gene" What has changed about the STAT3-SEAP construction that increases leakiness compared to the previous designs?

Reply:

As further discussed in our response to main point (1) of the same reviewer, we substituted the results in Figure 5 with those coming from transiently transfected receiver experiments. The previously observed reporter gene leakage in stable receivers could be explained by the preferential integration of the transgene within the bodies of active genes in the host cell's genome, by the lentiviral system.

(13) p19: "cells were transfected with 500 ng plasmid DNA ... for five hours" Is this correct?

Reply:

We transfected 2.4×10^5 cells plated the previous day in a 24-well with a total of 500 ng plasmid DNA using lipofectamine. Cells were incubated with the DNA-lipofectamine complex in OPTIMEM for 5 hours. Subsequently, the medium was refreshed to complete medium, containing the ligand. DNA plasmids were transfected in the following ratios; 11.9 receptor : 11.9 receptor : 1 STAT3 : 5.9 SEAP. Therefore, 193.8 ng per receptor dimer (387.6 ng when only one receptor was transfected), 96.1 ng STAT3-induced secreted alkaline phosphatase (SEAP) reporter plasmid pLS13 or pLS13-IL-10 and 16.3 ng STAT3 expression vector pLS15; adding up to 500 ng total plasmid DNA. However, in the case of the OR gate, a total of 693.8 ng DNA was transfected (193.8 ng per receptor dimer (3x) for a total of 581.4 ng, 96.1 ng STAT3-induced SEAP reporter plasmid pLS13 and 16.3 ng STAT3 expression vector pLS15). These experimental details are reported in the *Methods* section.

(14) The authors should explain the rationale for gating for single cells (FSC-A x FSC-H) shown in Fig S3.

Reply:

The FSC-A x FSC-H gate is constructed to discriminate single cells from duplets. This is denoted on the dot plots of the FSC-A x FSC-H. Additionally, we added the following text in the subsection “*Flow cytometry analysis and fluorescence-activated cell sorting*” of the *Methods* section: “As denoted in the figures, a first gate on FSC-A x SSC-A was constructed to identify the clonal population and a second gate on FSC-A x FSC-H was constructed to discriminate single cells from duplets.” (in yellow).

(15) Sequences of all new plasmids made in this study should be listed in the supplementary materials. Do the authors plan to deposit their plasmids/ cell lines in accessible repositories?

Reply:

Please find all novel plasmid sequences in the Supplementary information. Plasmids and cell lines are available upon reasonable request by the corresponding author. See the “*Availability of unique biological materials*” statement at the end of the main text (in yellow). Additionally, we plan on depositing our plasmids to Addgene after the work has been published.

(16) All source/ raw data should be made available as Supp data with the ms (journal guidelines permitting), or made available through a public repository.

Reply:

All datasets relating to the main and Supplementary Figures of this manuscript are now available via Zenodo at <https://zenodo.org/record/8055628> (see “*Data availability*” statement at the end of the main text, marked in yellow). Data include: SEAP activity and calibration curve values, ELISA values, Flow cytometry FCS files, raw blot and gel images, and raw grayscale fluorescence microscopy images. Additionally, all raw gel, blot, and grayscale microscopy images are provided as Supplementary Raw Data in the Supplementary Information document, in response to main point (5) of this reviewer (SS1-SS17).

Other points/ suggestions:

(17) Fig.1: There are too many diff colour schemes being used to depict the receptors, which is confusing. I would suggest that CC halves of a pair (e.g. A and A') and well as their ligand/s should be coloured using shades of the same colour.

Reply:

We thank the reviewer for this comment as we value clarity in our Figures and Images. We have colour-coded cognate CC pairs using shades of the same colour (A: light green, A': dark green; B: light red, B': dark red, Γ : light yellow, Γ' : dark yellow). All these changes have been implemented in all main and Supplementary Figures.

(18) Fig.2 : I think the point about CC directionality could be better made by including throughout an arrow in the CC symbol.

Reply:

We have included an arrow to indicate CC directionality in the main Figures.

(19) Fig2e: The symbols below the plot are a bit confusing. It may help to draw a horizontal line to group the concentrations 0 to 1 uM, indicating they are for the ligand dimer while only 2uM is for the monomer.

Reply:

We thank the reviewer for this suggestion and have incorporated the feedback in the Figure.

(20) Fig 2d: To ease understanding, please re-align the symbols below the plot such that all ligands are in the top row and all receptors are in the bottom row.

Reply:

The cartoons below the plot have been re-aligned according to the reviewer's suggestion.

(21) Fig 3c: As before, please re-align the symbols below the plot such that all ligands are in the top row and all receptors are in the bottom row. Also, to be less confusing please move the legend schematic, currently presented as plot inset, to the outside and put the labels I1, I2, I3 in the figure.

Reply:

The cartoons have been re-aligned according to the reviewer's suggestion.

(22) p19: "and 30oC (Figure 4" Missing information

Reply:

In the subsection "*Cell Culture and Transient Transfection*" of the *Methods*, the following information has been added: "*We opted to perform these experiments at 30°C, since the melting temperature (T_m) of the cognate B:B' and Γ : Γ' pairs has been reported to be around 40°C*" (marked in yellow).

(23) At several places in the methods section, references to supplementary figures/ tables seem to be incorrect. Please cross-check and fix them. A few examples below:
p18: "appropriate backbones for either transient or stable expression of the transgene (see Table S2)"

p19: "by titrating known concentrations of the hydrolysed pNPP product para-nitrophenol (pNP) (Supplementary Figure S1)"

p20: "A pET28a(+) vector (Supplementary Table S2) encoding"

Reply:

Reference to Figures and Supplementary Figures has been checked and corrected.

(24) p20: "stored at 80oC for subsequent use" Typo

Reply:

The typo has been corrected.

Reviewer #2 (Remarks to the Author):

Review

In this work, Pistikou et al. have engineered a scalable and orthogonal platform for synthetic communication in mammalian cells by re-purposing orthogonal coiled-coil (CC) interacting peptides for the GEMS (Generalized Extracellular Molecule Sensor) platform. Additionally, the authors engineered secreted ligands and used them as Boolean logic gate operators for cell communication. The paper is well-written and the data are clear and detailed. However, I have several concerns, most importantly regarding novelty.

Major comments:

(1) The novelty of this work is limited as the authors just combined two published platforms in a slightly modified way (CC interacting peptides from Chen et al, Nature, 2018: PMID: 30568301 and the GEMS platform from Scheller et al, Nat Chem Biol. 2018: PMID: 29686358). The main novelty here seems to be the engineering of the secreted ditopic ligands that stimulate their cognate receptors by optimizing different linkers along with SUMO tag. This could be considered an incremental improvement over prior work.

Reply:

We thank the reviewer for his time and effort reviewing our manuscript. We believe that CC functionalization of the GEMS receptor platform consists of a novel technology for synthetic biology applications. In particular, we would like to highlight that we indeed combine the CC technology (Gradišar et al., 2010, *J Pept Sci*, 10.1002/psc.1331 and Lebar et al., 2020, *Nat Chem Biol*, 10.1038/s41589-019-0443-y) with the GEMS platform developed by Scheller et al. (2018, *Nat Chem Biol*, 10.1038/s41589-018-0046-z) to design a novel synthetic communication system in mammalian cells. To enable synthetic communication in mammalian cells, we design a SUMO-tagged engineered dipeptide ligand that can be secreted by a sender cell population. We thank the reviewer for noting the novelty of this approach in his comment. This point is now stated in paragraph #2 of the *Introduction* section (marked in yellow): “To enable synthetic communication between populations of mammalian cells, we design a novel small ubiquitin-like modifier (SUMO) tag-fused CC dipeptide that can be secreted by mammalian cells.” An important advantage of the CC-GEMS platform over other technologies that were used to engineer synthetic communication in mammalian cells is that it has the potential to be highly scalable, as programmable design of large sets of CC heterodimers has

previously been achieved (Chen et al., 2019, Nature, 10.1038/s41586-018-0802-y). This is noted in the *Introduction*: “Programmable design of large sets of CC heterodimers has been previously achieved (Chen et al., 2019, Nature, 10.1038/s41586-018-0802-y), providing the platform with potential scalability.” (paragraph #2, in yellow). Next to showing CC-GEMS mediated synthetic communication, we demonstrate that the platform is capable of performing distributed Boolean logic operations at the receptor level, which is also novel (see Figures 4 and 5). In the first paragraph of the introduction, we also note “...approaches utilizing small molecules have managed to engineer mammalian cells so they are capable of complex bio-computations (Ausländer et al., 2018, Nat Methods, 10.1038/nmeth.4505). However, these approaches lack in scalability due to the use of small molecules as signal initiators and cannot perform Boolean logic operations at the receptor level.”

(2) Several groups have developed more robust and clinically relevant systems combining orthogonal ligands with cognate receptors in immune cells. This work has not even been cited, which would have exposed the limited novelty. In principle, these prior systems can be used in a similar way as presented in this work because they are orthogonal, secreted, and even immune-compatible (Kalbasi et al., Nature, 2022: PMID: 35676488; Sockolosky et al., Science, 2018, PMID:29496879; Zhang et al., Sci Transl Med, 202 : PMID: 34936380).

Reply:

We would like to thank the reviewer for this insightful comment. Although the works of Kalbasi et al. (2022), Sockolosky et al. (2018), and Zhang et al. (2022) do utilize orthogonal, secreted ligands that are immune compatible, and can in principle be used for construction of a mammalian synthetic communication system, we do believe they differ from our developed technology in several ways. We would like to point to the reviewer that the work of Sockolosky et al. (2018) has been cited in the original manuscript (ref #38; “...in addition to approaches that employ directed evolution of naturally occurring polypeptides.”), however the more recent works of Kalbasi et al. (2022) and Zhang et al. (2022) were not cited in the original document. We have now added those citations in the revised manuscript (ref #39 and #40 in yellow). As we have pointed out in the *Introduction*, we believe that a main difference between the cited work and ours is that they use labour intensive methods (site directed evolution) to render the input signal orthogonal, limiting scalability. In contrast, our platform uses a “plug and play” approach of previously designed parts (CCs). As pointed in the *Introduction*, “Programmable design of large sets of CC heterodimers has previously been achieved...” (Chen et al., 2019, Nature, 10.1038/s41586-018-0802-y), providing the platform with potential scalability. Additionally, although a set of chimeric receptors fusing an orthogonal IL-2 extracellular domain (ECD)

to the γ_c -chain (γ_c) of a set of cytokines have been engineered, such an approach cannot achieve Boolean operations at the receptor level and is fundamentally limited to the corresponding γ_c -cytokine-related signal. We have now modified the *Introduction* to highlight the abovementioned points: “Directed evolution of natural proteins consists of a labour-intensive method to render input signals orthogonal from each other, while in addition such an approach lacks the inherent ability to institute Boolean logic. Furthermore, although a set of chimeric receptors have been engineered using an orthogonal interleukin 2 (IL-2) extracellular domain (ECD) to the γ_c -chain (γ_c) of a set of cytokines, the elicited response is limited to the corresponding γ_c cytokine-related signal.” (in yellow). Lastly, we would like to point out that CCs have been shown to be immune compatible (Ljubetič et al., 2017, *Nat Biotechnol*, 10.1038/nbt.3994) and as such we do not expect immune-related issues when CC-GEMS is used in an *in vivo* setting. This point has been added in the *Conclusion* section (paragraph #4; in yellow): “Since the immune compatibility of CCs has been previously shown (Ljubetič et al., 2017, *Nat Biotechnol*, 10.1038/nbt.3994) when CC protein-origamis were assessed in mice, we similarly expect no innate immune response upon introduction of CC-GEMS *in vivo*”.

(3) The author stated that this platform can be utilized in some applications, but a compelling application is missing as is convincing proof, that the system is superior to prior work. The Boolean logic gate is not a compelling application and it can be achieved in many other ways using existing tools, even for cell communication. For example, Fink et al, *Nat Chem Biol*. 2019: PMID: 30531965 designed a full set of orthogonal Boolean logic gates based CC interacting peptides; Again, this manuscript lacks novelty.

Reply:

To show the potential of CC-GEMS to be used in a therapeutic setting, we decided to include additional experiments to demonstrate that CC-GEMS can control the secretion of a therapeutic protein. In our revised manuscript, we show that CC-GEMS can control the expression of IL-10 (Figure 6). In more detail, we supplemented the *Results and Discussion* section with a subsection titled “CC-GEMS can control secretion of therapeutic ligands”, where we state: “To demonstrate the potential of CC-GEMS to be used for therapeutic purposes, we re-engineered the system to secrete a therapeutic protein as a response to cognate ditopic CC ligand... HEK293T cells were transiently transfected to express A-type receptor as well as STAT3 and interleukin-10 (IL-10) under the control of a STAT3 recognition element... only upon addition of cognate ligand, IL-10 is secreted... These results demonstrate that CC-GEMS can secrete therapeutic proteins upon addition of cognate ligands.”

Regarding the Boolean logic operations, we argue that there are fundamental differences in the SPOC system, designed by Fink et al. (2019) and CC-GEMS. Although the authors demonstrate Boolean logic based on CC pairs, the system still relies on small molecules as signal initiators, limiting scalability at the input signal level. Additionally, although the authors demonstrate multilayered signaling within the cell, the SPOC system does not permit multilayered signaling between cell populations, as the split protease-CC fusion protein cannot diffuse between different cell populations. To address this point in the revised manuscript, in paragraph #2 of the *Introduction* we mention: “*Although CC-mediated Boolean gates have been engineered in mammalian cells at the intracellular level (Fink et al., 2019, Nat Chem Biol, 10.1038/s41589-018-0181-6), they have not been used to control receptor activation and enable intercellular communication.*” (in yellow).

(4) In the GEMS publication, the authors used several receptors that activate STAT3, NF-kb, NFAT, or MAPK pathways. Is there any reason why the authors focused on the STAT3 pathway. Is their system incompatible with the other signalling pathways?

Reply:

We would like to thank the reviewer for this comment, since we believe it is important that it becomes clear in the manuscript that all endogenous pathways can be rewired to produce a custom response using CC-GEMS, adding to the platform’s modularity. Here, we provide proof of principle by using the JAK-STAT pathway, however both the MAPK and PI3K/Akt pathways can in principle be used in a CC-GEMS setting. We have highlighted this in paragraph #1 of the *Conclusion*: “*...although we utilize here the rewired JAK-STAT pathway for CC-GEMS construction, both MAPK and PI3K/Akt used in the original GEMS platform (Scheller et al., 2018, Nat Chem Biol, 10.1038/s41589-018-0046-z) can also be used, equipping CC-GEMS with additional, output-related modularity.*”

(5) In figure 4, the author showed bitopic activators that stimulate the receptors. What is the advantage of these ligands over Sun-Tag-mCherry and PSA (prostate-specific antigen) that are shown in the original GEMS platform? In both cases, these proteins can be secreted and can be used as described in Fig 5. Using nanobodies is also scalable, even more than CC interacting peptides.

Reply:

The major advantage of CC-GEMS over the original GEMS platform relates to the capacity of CC-GEMS to display Boolean logic (see Fig 4b-d and 5d) at the receptor level without introducing multiple receptor types. Theoretically, AND gate logic can be achieved using a set of different GEMS receptors

in a manner similar as to how CARs and SynNotch receptors were used previously (Choe et al., 2021; *Sci Transl Med*, 10.1126/scitranslmed.abe7378 and Hyrenius-Wittsten et al, 2021; *Sci Transl Med*, 10.1126/scitranslmed.abd8836). However, such transcription-dependent AND gate logic results in a delayed response due to controlled expression of the second receptor compared to direct AND-based activation at the receptor level enabled by CC-GEMS (see Fink et al., 2019; *Nat Chem Biol*, 10.1038/s41589-018-0181-6). Additionally, in our design, AND gate logic is attainable by expressing a single CC-GEMS receptor, while the transcription-dependent AND gate logic described above requires the expression of two sets of receptors, introducing additional genomic burden to the cell. We have now supplemented the *Conclusion* section of our revised manuscript with the following point of discussion *“Theoretically, AND gate logic could be achieved using a set of original GEMS receptors responding to diverse ligands in a manner similar to CAR and SynNotch receptors were previously used (Choe et al., 2021; Sci Transl Med, 10.1126/scitranslmed.abe7378 and Hyrenius-Wittsten et al, 2021; Sci Transl Med, 10.1126/scitranslmed.abd8836), where one GEMS detects a cognate ligand and subsequently triggers the expression of a second GEMS that can be activated by a second ligand, producing the final output. However, such transcription-dependent logic introduces additional burden to the cell due to the introduction of two distinct receptor types and results in a delayed response. In contrast, AND gate logic in CC-GEMS occurs at the receptor level and can be engineered using a single CC-GEMS receptor type.”* (paragraph #3, in yellow).

(6) If the goal is to engineer secreted ligands as presented in figures 4 and 5, what is the purpose of figure 2 where the authors artificially synthesized a ditopic ligand using BM(PEG)3?

Reply:

Figure 2 serves as proof of principle that A-type receptors can be selectively activated by soluble, A'-A' ditopic ligands. Since A and A' CCs bind in a parallel manner, we started by engineering A'-A' dipeptides that are N' terminally linked and assessed their capacity to elicit a receptor response. At the beginning of paragraph #1 of the subsection *“Design of soluble ditopic CC ligands to activate CC-GEMS receptors”* of the *Results and Discussion*, we state: *“...we next aimed to assess if CC-GEMS receptor dimerization and activation can be achieved by soluble, synthetic, CC ligand dipeptides. Considering the parallel orientation of bound CC cognate pairs (Gradišar et al., 2010, J Pept Sci, 10.1002/psc.1331), we engineered A'-A' dipeptides that are N-termini linked”* (in yellow). When we established that this is the case, we reasoned that a longer linker spanning the N-terminus of one CC to the C-terminus of another would be able to facilitate proper CC orientation to activate cognate

receptors. This work, along with a study relating to linker length, is presented in Figure 3 of the manuscript.

(7) While receptor activation is orthogonal, the STAT3 pathway is still endogenous and it can be activated by multiple physiological clues. This is especially true when applying this system in a therapeutic context.

Reply:

It is indeed the case that CC-GEMS, together with the original GEMS platform, uses rewired, endogenous pathways for synthetic receptor activation. For instance, the JAK-STAT pathway (used in the present work) can be activated by an array of endogenous ligands that can potentially crosstalk to CC-GEMS. In particular, STAT3 phosphorylation can happen via IL-6/IL6R, IL10/IL10R, or IL23/IL23R interactions. Cells lacking corresponding receptors or expressing them at low levels can be used for synthetic receptor engineering, minimizing anticipated crosstalk. Additionally, as we argue in paragraph #4 of the *Conclusion*: *“Future research could focus on engineering alternative dimeric synthetic receptors to respond to CC modules. For instance, MESA (Modular Extracellular Sensor Architecture) (Daringer et al., 2014, ACS Synth Biol, 10.1021/sb400128g) or DocTAR (Double-cut Transcription Activation Receptor) (Zhou et al., 2023, Cell Rep, 10.1016/j.celrep.2023.112385) receptors could similarly be re-engineered to express CCs as their extracellular domain that dimerize upon the addition of ditopic CC ligands. Such an approach has the potential to yield a maximally orthogonal communication system based on synthetic receptors, since in the MESA and DocTAR architecture, signaling downstream from the engineered receptors is not subject to cross-talk with native cellular pathways.”* (in yellow).

Minor comments:

(8) Why do the authors normalize SEAP levels? Absolute values are needed to judge the leakiness of the system.

Reply:

We agree with the reviewer and as mentioned above (response in comment #3, #8, and #11 of reviewer 1) we opted for showing the non-normalised versions of the data. Please find all new Figures and Supplementary Figures showing non-normalised values in the revised manuscript.

(9) The system presented in figure 5b seems to be very leaky (SEAP units) and the performance is really

poor (less than two fold change between the middle columns- with and without doxycycline). This is not a convincing performance.

Reply:

We agree with the reviewer and for that reason, we undertook experiments to improve the system's performance. These new data are now presented in Figure 5. As the reviewer states, the system presented in Figure 5b in the original manuscript is very leaky resulting in poor performance (less than two-fold). We therefore focused on resolving that issue by engineering receiver cells, where the genes of interest were transiently introduced (see response to reviewer 1). As depicted in Figure 5b, the system's performance has been improved 3.5-fold. In paragraph #2 of the subsection "*Establishing intercellular communication using CC-GEMS*" of the *Results and Discussion*, we therefore state: "...we observe a 3.5-fold increase in SEAP activity for receiver cells cultured with senders in the presence of dox compared to a co-culture with senders in the absence of the inducer." (in yellow).

(10) In figure 5d, the comparison should be between cells co-cultured in the presence or absence of doxycycline and not between different sender cell types.

Reply:

For this experimental design, we were not interested in introducing the control of a chemical switch and constructed cells that maximally express the ligands even in the absence of doxycycline. In that case, the true negative control consists of the receiver population incubated with control sender cells that do not produce ligand.

(11) This reviewer does not understand why SEAP activity suddenly converted to substrate cleavage in 1c and why you need a time course measurement. In fact, this figure is included in figures 1d and 1a.

Reply:

For an answer to this comment, we direct the reviewer to the subsection "*SEAP quantification*" of the *Methods*, where we state that: "*SEAP concentration in cell culture medium was quantified in terms of absorbance increase due to para-nitrophenyl phosphate (pNPP), using a commercially available SEAP assay kit... Absorbance values were measured at 405 nm, at a controlled temperature of 25°C, for 60 min.*" This is a kinetic measurement and is depicted on Figure 1c. We then proceeded to determine sample SEAP concentration in units per litre (U/L), where absorbance units are converted to amount

of substrate conversion and SEAP activity is calculated from the slope of the time trace, using a calibration curve.

(12) Methods: All reagents used should be exactly cited along with the company name and catalog number.

Reply:

Reagents and materials are reported together with the company name and catalogue number in the Methods section.

Reviewers' Comments:

Reviewer #1:

Remarks to the Author:

The authors have addressed several points raised, mostly satisfactorily. However, I do have some remaining concerns.

(1) Switching from stably to transiently transfected receivers improves the results presented in Fig 5. As a way of reducing output leakiness, this is understandable. But, I wonder in which therapeutic/ diagnostic setting would transiently modified receivers be relevant?

(2) Please indicate in each relevant figure legend that the 3 repeats were done on the same day.

(3) "endogenous receptor dimerization occurs ... outcomes of linker length in receptor activation". This explanation should be supported by literature evidence.

(4) Fig 2d: not sure why the a1 p-value (rather than ns) is in the figure, when the rest are not.

(5) I thank the authors for promising to deposit their plasmids to Addgene after publication. However, I would really encourage them to do so before publication so that they can include the accession numbers in the paper. It is possible to deposit plasmids to Addgene and obtain accession numbers, while requesting to make them publicly available only after publication.

Reviewer #2:

Remarks to the Author:

In the revised version of the manuscript, Pistikou et al. performed additional experiments, modified the text, and cited relevant and important work in the field. However, the author has only partially addressed the limited scientific novelty of this work. In addition, the overall performance of the system remains modest, especially for the cell-cell communication experiments (Figure 5 b, d where the fold change is mainly around 3). In the revised version, the authors stated that "Notably, although we utilise here the rewired JAK-STAT pathway for CC-GEMS construction, both MAPK and PI3K/Akt used in the original GEMS platform can also be used, equipping CC-GEMS with additional, output-related modularity". However, the author did not show any experiment that supports this claim.

REVIEWERS COMMENTS AND AUTHOR'S ANSWERS

Reviewer #1 (Remarks to the Author):

The authors have addressed several points raised, mostly satisfactorily. However, I do have some remaining concerns.

(1) Switching from stably to transiently transfected receivers improves the results presented in Fig 5. As a way of reducing output leakiness, this is understandable. But I wonder in which therapeutic/diagnostic setting would transiently modified receivers be relevant?

Reply:

We thank the reviewer for this comment and appreciate their concern. Indeed, we believe that in order to be able to translate CC-GEMS in a therapeutic setting, actions must be taken to stably introduce the relevant genes in the cell genome. However, that is a laborious and long process that we believe is beyond the scope of the present work, as here we only aim to provide proof of concept that CC-GEMS functions in mammalian cells. We would like to point out that most published work in the field, including the development of the original GEMS platform (Scheller, 2018, *Nat Chem Biol*; 10.1038/s41589-018-0046-z), uses transient transfection methods.

These papers include:

- The development of the MESA synthetic receptor (Daringer, 2014, *ACS Synth Biol*; 10.1021/sb400128g)
- The development of the Tango synthetic receptor (Barnea, 2008, *PNAS*; 10.1073/pnas.0710487105)
- The development of a dCas9-based synthetic receptor (Baeumler, 2017, *Cell Rep*; 10.1016/j.celrep.2017.08.044)
- The development of GEARS (Krawczyk, 2020, *Nat Commun*; 10.1038/s41467-020-14397-8)
- The development of the DoctAR synthetic receptor (Zhou, 2023, *Cell Rep*; 10.1016/j.celrep.2023.112385)
- The development of the OCARs (Mahameed, 2022, *Nat Commun*; 10.1038/s41467-022-35161-0)
- The establishment of synthetic communication in mammalian cells using L-tryptophan (Bacchus, 2012, *Nat Biotech*; doi.org/10.1038/nbt.2351)
- The establishment of synthetic communication in mammalian cells via L-tryptophan and IL-4 (Kolar, 2015, *BMC Syst Biol*; 10.1186/s12918-015-0252-1)
- The construction of synthetic communication in mammalian cells using NO (Wang, 2008, *Exp Cell Res*; 10.1016/j.yexcr.2007.11.023)
- The establishment of full-adder computations in mammalian cells using small molecules (Ausländer, 2018, *Nat Methods*; 10.1038/nmeth.4505)

Future work could focus on stably integrating CC-GEMS into the cell genome. For this purpose, several steps for lentiviral transfection optimization and control of reporter gene leakage can be taken; i.e. titrating the lentivirus to the cells and screening monoclonal cell lines for optimal performance.

(2) Please indicate in each relevant figure legend that the 3 repeats were done on the same day.

Reply:

In response to this remark, the fact that the repeats were done on the same day is now stated in the figure legend (see changes marked in yellow in the manuscript and supplementary information).

(3) "endogenous receptor dimerization occurs ... outcomes of linker length in receptor activation". This explanation should be supported by literature evidence.

Reply:

We have supplemented the main text by referencing a paper (ref #55, in yellow; Bethani, 2010, *EMBO J*; 10.1038/emboj.2010.175) where the authors reviewed the literature regarding the influence of membrane organization on receptor activation.

(4) Fig 2d: not sure why the a1 p-value (rather than ns) is in the figure, when the rest are not.

Reply:

We thank the reviewer for noting this. We have now changed the figure showing *ns* above the fold change. The p-value (nearing significance) is now reported on Supplementary Table S6.

(5) I thank the authors for promising to deposit their plasmids to Addgene after publication. However, I would really encourage them to do so before publication so that they can include the accession numbers in the paper. It is possible to deposit plasmids to Addgene and obtain accession numbers, while requesting to make them publicly available only after publication.

Reply:

We thank the reviewer for his suggestion. We have started the procedure on depositing our plasmids to Addgene and plasmids will be available after publication. All the accession numbers of the deposited plasmids are now marked in yellow.

Reviewer #2 (Remarks to the Author):

In the revised version of the manuscript, Pistikou et al. performed additional experiments, modified the text, and cited relevant and important work in the field. However, the author has only partially addressed the limited scientific novelty of this work. In addition, the overall performance of the system remains modest, especially for the cell-cell communication experiments (Figure 5 b, d where the fold change is mainly around 3). In the revised version, the authors stated that “Notably, although we utilise here the rewired JAK-STAT pathway for CC-GEMS construction, both MAPK and PI3K/Akt used in the original GEMS platform can also be used, equipping CC-GEMS with additional, output-related modularity”. However, the author did not show any experiment that supports this claim.

Reply:

We would like to thank reviewer 2 for their comments, since following their advice, we have supplemented the manuscript with additional experiments that improved the overall performance of the system and experiments that prove that CC-GEMS can be used in the context of two distinct, alternative pathways.

We would like to highlight once more that the novelty of our work lies on the fact that CC-GEMS can achieve highly *scalable* and *programmable* synthetic, intercellular communication in mammalian cells *at the receptor level*.

As mentioned above, in the revised manuscript, we managed to improve the performance of the system for the sender-receiver experiments (previously Figure 5, now Figure 6). In detail, regarding Figure 6b (previously Figure 5b), we show *6.6-fold* increase in SEAP activity for receiver cells cultured with senders in the presence of dox compared to a co-culture with senders in the absence of the inducer and *9.7-fold* increase compared to receiver cells cultured with control cells. This increase in performance happens primarily due to resolving ligand-related leakage problems in the sender cell population (see Supplementary Figure S28). More specifically, we screened monoclonal cell lines for clones that minimally express ligand in the absence of the inducer and respond to dox by expressing ligand in high amounts (see in particular Supplementary Figure S28d). These cells were used for the construction of a dox-induced sender-receiver experiment and resulted in the response depicted in Figure 6b. In addition, the performance of the AND gate in Figure 6d (previously Figure 5d) was enhanced by increasing the surface area available for cells to be seeded. We did that by utilising well-plate inserts. In the “Methods” section of the revised manuscript, we note: “*To increase the surface area available to cells seeded, we utilised Millicell Cell Culture Inserts (3.0 μm pore, translucent PET membrane; PTSP24H48, Merck). In detail, we seeded 3 x 10⁵ sender 1 and 3 x 10⁵ sender 2 or control HEK293S GnTi- on the well-plate surface, as well as 2 x 10⁵ sender 1 and 2 x 10⁵ sender 2 or control*

HEK293S GnTi- on the insert (in yellow). This optimization step resulted in, *7-fold* increase in reporter gene expression compared to receiver cells incubated with sender population 1, *9.6-fold* increase in reporter gene expression compared to receiver cells incubated with sender population 2, and *19.3-fold* increase in reporter gene expression compared to receiver cells incubated with control. As we also note in the revised manuscript, this performance is akin to performances seen in previously engineered paracrine synthetic communication systems, ranging from 1.5-20-fold increase. The relevant literature is cited in the main text.

Lastly, in the revised manuscript, we have performed additional experiments where we show that CC-GEMS is compatible with the PLCG pathway, besides JAK/STAT. Specifically, we supplemented the main text with an additional Figure (Figure 4 in the revised manuscript), where we show SUMO-tagged A'-A' ligand-dependent activation of cells transfected with A'-type_{PLCG} receptor and NFAT-induced secreted alkaline phosphatase (SEAP) reporter plasmids. Please find these changes under the subsection entitled: *"Rerouting CC-GEMS Signalling through Alternative PLCG Pathway Activation"* (in yellow).

Reviewers' Comments:

Reviewer #1:

Remarks to the Author:

I thank the authors for satisfactorily addressing all my concerns.

Reviewer #2:

Remarks to the Author:

The authors have convincingly addressed all of my and my fellow reviewers concerns

REVIEWERS COMMENTS AND AUTHORS' ANSWERS

Reviewer #1 (Remarks to the Author):

I thank the authors for satisfactorily addressing all my concerns.

Reviewer #2 (Remarks to the Author):

The authors have convincingly addressed all of my and my fellow reviewers concerns.

Authors' response

We would like to express our sincere gratitude to both reviewers for their invaluable feedback, which we firmly believe significantly contributed to improve the final version of the paper.